# A self-assembled protein β-helix as a self-contained biofunctional motif

Camilla Dondi[1,2], Javier Garcia-Ruiz [1,3], Erol Hasan [1,4], Stephanie Rey [1], James E. Noble [1], Alex Hoose[1], Andrea Briones[1], Ibolya E. Kepiro[1], Nilofar Faruqui[1], Purnank Aggarwal [1], Poonam Ghai[1], Michael Shaw [1,5], Antony T. Fry[1], Antony Maxwell[1], Bart W. Hoogenboom [2,6], Christian D. Lorenz [3] & Maxim G. Ryadnov [1,3]✉

Nature constructs matter by employing protein folding motifs, many of which have been synthetically reconstituted to exploit function. A less understood motif whose structure-function relationships remain unexploited is formed by parallel β-strands arranged in a helical repetitive pattern, termed a β-helix. Herein we reconstitute a protein β-helix by design and endow it with biological function. Unlike β-helical proteins, which are contiguous covalent structures, this β-helix self-assembles from an elementary sequence of 18 amino acids. Using a combination of experimental and computational methods, we demonstrate that the resulting assemblies are discrete cylindrical structures exhibiting conserved dimensions at the nanoscale. We provide evidence for the structures to form a carpet-like three-dimensional scaffold promoting and inhibiting the growth of human and bacterial cells, respectively, while being able to mediate intracellular gene delivery. The study introduces a self-assembled β-helix as a self-contained bio- and multi-functional motif for exploring and exploiting mechanistic biology.

Nature employs tandem repeats in amino-acid sequences to construct symmetric structures ranging from enzymatic complexes to protein filaments and viral capsids[1,2]. Such repeats fold into super-secondary structure motifs, many of which have been synthetically reconstituted and optimised to generate functional ensembles across length scales[3,4]. More motifs become available through the systemic prediction and analysis of protein folds[5,6]. A relatively less understood motif, which remains unexploited, is formed by parallel β-strands associating in a helical repetitive pattern that is originally termed a β-helix[7]. Recent searches revealed that over 340 native polypeptide chains comprise β-helical domains[8]. Lyases, transferases, anhydrases and antifreeze proteins are among common examples of β-helix proteins[7–12]. The motif has also been shown to form amyloid-like spines thus representing a structural element that bridges globular and fibrous proteins[12–14].

Known β-helices are elongated prisms or ladders built of two- or three-faced units or rungs[7,13–15]. The prisms can be right- or left-handed. However, triangular units consecutively coiling into left-handed helices appear to dominate[16]. These units exhibit hexapeptide repeats [LIV]-[GAED]-X$_2$-[STAV]-X or [LIV]-G-X$_4$ that are conserved among protein structures sourced from PROSITE (e.g., PDOC00094)[17], Pfam (e.g., Clan CL0536)[18], DbStRiPs (e.g., LbH)[8], RepeatsDB v3.0 (e.g., β-solenoid)[1], and AlphaFold (e.g., protein structure database)[19]. Three consecutive hexapeptide repeats forming three β-strands separated by turns constitute one equilateral triangular coil[7–14]. A network of interactions between similar residues in consecutive units or coils stabilise the resulting β-helix. Apart from hydrophobic contacts, which run in and through the plane of the triangles, hydrogen bonds between asparagine residues furnish asparagine ladders, while aromatic (Trp)

[1]National Physical Laboratory, Teddington, UK. [2]London Centre for Nanotechnology, University College London, London, UK. [3]Department of Physics, King's College London, London, UK. [4]Division of Physical Science and Engineering, King Abdullah University of Science and Technology, Thuwal, Saudi Arabia. [5]Department of Computer Science, University College London, London, UK. [6]Department of Physics & Astronomy, University College London, London, UK. ✉e-mail: max.ryadnov@npl.co.uk

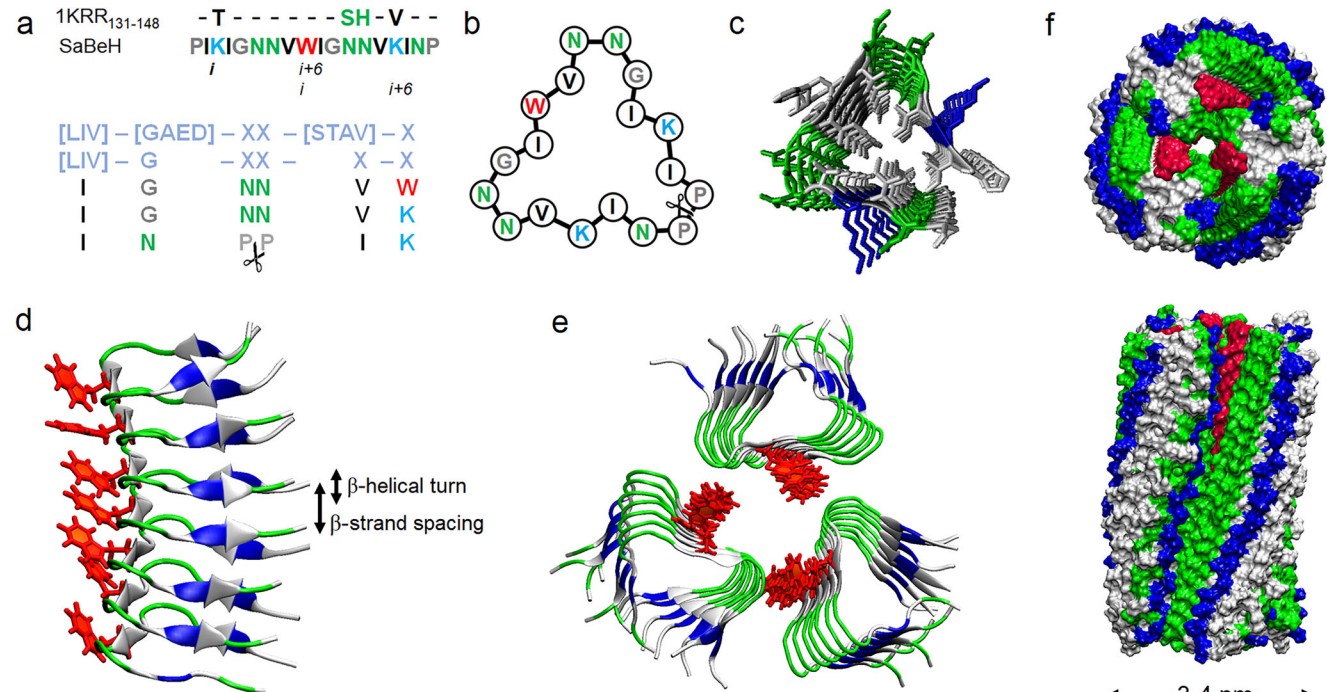

**Fig. 1 | β-helix design. a** Amino-acid sequence of SaBeH aligned with the sequence of 1KRR$_{131-148}$ (upper), with hexapeptide repeats characteristic of β-helical domains (lower), and **b** as configured into a left-handed β-helical rung with each side representing a β-strand. Repeating residues in a hexapeptide repeat (*i, i + 6*) are highlighted for a lysine ladder. Colour coding: black and blue denote hydrophobic and cationic residues, proline and glycine are in grey, tryptophan is in red, asparagine is in green. **c** A top-down schematic of the assembly. For clarity only asparagine (green) and lysine (blue) are shown in colour. **d** Side view of MD simulation performed over 100 ns for a β-helix composed of seven β-helical turns and **e** top-down view of MD simulation performed over 100 ns for a β-helical trimer. Arginine and lysine residues and tryptophan side chains are shown in green, blue, and red, respectively. **f** Top-down (upper) and side (lower) views of MD simulation performed over 100 ns for a 12-turn β-helical trimer. Colour coding is as in (**e**).

and aliphatic (Pro) rings stack together. The simplest β-helix is postulated to comprise two coils connected via a short loop formed by glycine or proline residues[20]. This arrangement is compatible with the estimates made by mass-per-length measurements for a contiguous polypeptide chain folding into a β-helix. Namely, each molecule spans two β-helical turns which constitutes half a molecule per spacing between two β-strands at $0.47 \pm 0.01$ nm[21,22]. We reason that one triangular unit comprising 18 residues is sufficient to assemble into an extended β-helix in a non-covalent fashion. Such a coil unit formed by an open peptide chain, with both N-and C-termini free, retains the axial rise of 0.47 nm, supporting the coil-like stacking of individual units without covalently connecting them. We further reason that, since the rungs are nearly planar, their assembly is secured by parallel hydrogen-bonded β-strands and a shared hydrophobic core running through all the units. Finally, with one of the three faces in triangular prisms capable of providing an extended binding face promoting triple-helix formation, a stable tubular structure can be generated with pre-defined surface characteristics supporting specialist biology. Based on these conventions, we introduce herein a self-assembling β-helix. The helix assembles from an elementary coil unit, a linear polypeptide chain of 18 amino acids, designed to form discrete cylindrical structures. We demonstrate that such structures promote and inhibit the growth of human and bacterial cells, respectively, support three-dimensional (3D) human cell culture whilst mediating gene transfer in 2D and 3D thus introducing a self-contained, multifunctional motif.

## Results
### β-helix design
The elementary unit design derives from a β-helical region of galactoside acetyltransferase[23]. A one-turn unit comprising residues 131–148

(PDB: 1KRR) exhibiting characteristic hexapeptide repeats was identified as a basic rung repeat in the helix (Fig. 1a).

The unit forms an equilateral triangle, in which hydrophobic residues point inwards stabilising the unit. Other characteristic features include asparagine pairs supporting intra-helical asparagine ladders, glycine and asparagine residues forming two complete turns connecting three β-strands, and a solvent exposed tryptophan residue promoting inter-helix contacts. The unit contains another turn starting with an asparagine followed by a C-terminal proline. In the wild-type protein this turn, completed with a glycine, connects with the consecutive coil[20,23]. In the designed unit, the C-terminal and N-terminal proline residues non-covalently abut (Fig. 1b). With the other two turns fixing the conformation of the unit, such a "non-covalent" turn introduces a degree of flexibility allowing the free termini to accommodate the axial rise upon β-helix trimerization. However, it also eliminates covalent bonding between consecutive rungs. To mitigate this, the two complete turns were made GNN promoting the formation of two axial asparagine ladders (Fig. 1c). In addition, glycine and asparagine residues have the highest propensities for β-turns among amino acids[24], further stabilising the triangular fold. To maintain discrete hydrophobic and polar faces of the triangle, the wild-type pattern of alternating hydrophobic and polar tripeptides was disrupted by two lysine residues introduced into two hydrophobic tripeptides on two sides of the triangle (Fig. 1a–c). This arrangement ensures that inter-helical contacts are driven by one side of the unit incorporating a single tryptophan residue which was kept solvent exposed (Fig. 1d). The other two sides of the rung become positively charged whereas all the hydrophobic contacts are confined to the core of the unit and consequently of the helix. The two lysine residues, together with tryptophan, make the helix capable of binding to membranes, in particular neoplastic and bacterial, which are anionic. The rigid geometry of the fold

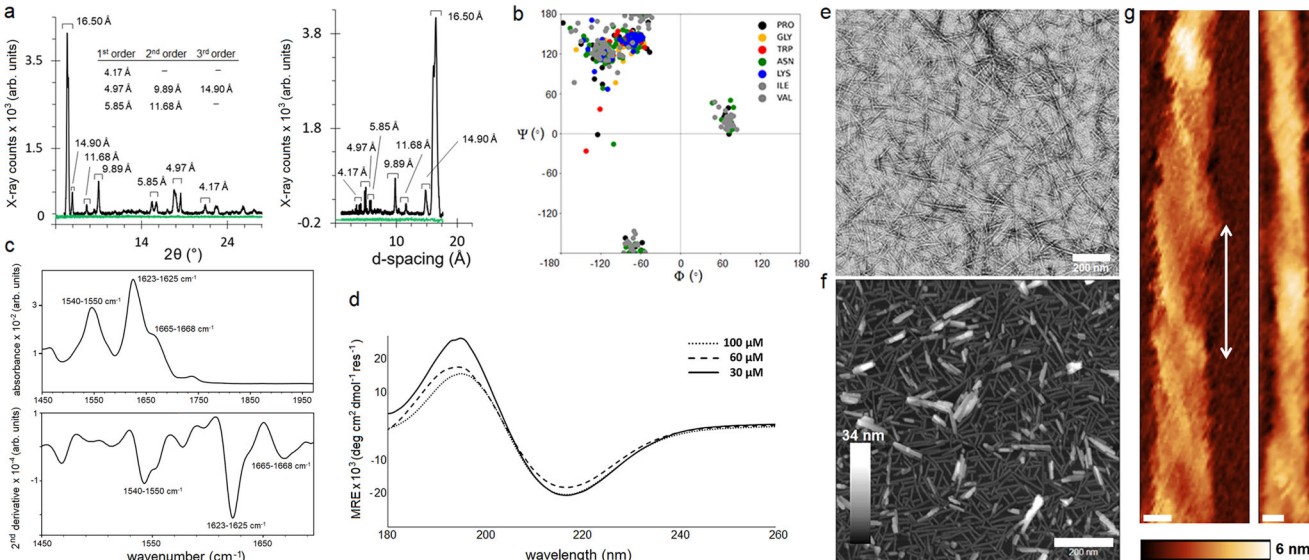

**Fig. 2 | β-helix folding and assembly. a** XRD patterns for SaBeH (black) and a non-assembly control peptide (green). X-ray counts are shown versus both 2θ (left) and d-spacings (right). **b** Ramachandran plot for SaBeH as in Fig. 1f. **c** FT-IR spectrum (upper) and its 2nd derivative (lower) for SaBeH. **d** CD spectra for SaBeH at different concentrations. **e** Electron and **f** atomic force micrographs of SaBeH. Scale bars are 200 nm. Colour (height) scale bar is 34 nm. **g** Atomic force micrographs of individual SaBeH cylinders highlighting a left-handed twist with a pitch of 28.5 ± 7.3 nm indicated by a double-headed arrow. Scale bars are 5 nm. Colour (height) scale bar is 6 nm. Assembly conditions: 100 μM (unless stated otherwise) in 10 mM 3-(Morpholin-4-yl)propane-1-sulfonic acid (MOPS), pH 7.4, at room temperature. Images in (**e**–**g**) are representative of at least 3 independent experiments.

limits the ability of individual peptide units to bind to and re-fold in the membranes. However, upon the assembly of the units into a β-helix, lysine residues build up contiguous cationic ladders or faces with high affinity to the membranes (Fig. 1e, f)[25]. Upon membrane binding inter-helical contacts are destabilised exposing tryptophans to interact with lipids. Given that sequences with the proportion of hydrophobic residues exceeding 50% elicit haemolytic and cytotoxic effects, hydrophobic to non-hydrophobic residues in the unit were kept at the ratio of 1:1.5, allowing for non-toxic membrane-binding and cell-penetrating activities[26].

Figure 1 sums up the design. The elementary unit is an equilateral triangle folded into three β-strands linked via two covalent and one non-covalent turns (Fig. 1a–c and Supplementary Fig. 1). The terminal prolines vacillate, allowing to both support the assembly by stacking their aliphatic rings and enhance its destabilisation caused by membrane binding. Multiple copies of the unit assemble into a β-helix via continuous in-register parallel β-sheets. The β-helix trimerizes via its tryptophan side giving rise to a highly cationic cylindrical structure. With each side of the triangular unit spanning ~2 nm (0.33–0.35 nm per residue, six residues per side), the diameter of the cylinder is estimated to be ~3.5 nm. This is assuming a perfect equilateral triangle formed by three triangular units (Fig. 1e). Considering irregular spacings between β-helices the diameter is expected to vary between 3 and 4 nm (Fig. 1d–f).

## β-helix folding and assembly

Atomistic molecular dynamics (MD) simulations supported the estimated dimensions of the assembly. Inter-molecular distances between Cα-atoms of amino-acid residues at the opposite ends of the β-helical trimer as shown in Fig. 1e were consistent with the estimated diameter (Supplementary Fig. 2). The estimated distances were practically identical for 100-ns and 200-ns simulations confirming that these distances were conserved (Supplementary Figs. 2–5). Specifically, average distances between prolines of different helices in each β-helical turn were 4-5 nm, while the C-terminal asparagines were separated from the GNN turns at ~3.5 nm (Supplementary Fig. 2). The

shortest distances, 1–1.2 nm, were recorded between W9 and N7 of different helices. Adding the size contributions of the side chains this indicates amide-π interactions that typically occur at 0.4–0.7 nm[27]. Other inter-helical interactions also clustered around W9 (Supplementary Fig. 2). These findings suggest that the interface between β-helices is formed by W9 and N6N7 ladders, which is in close agreement with the mode of inter-helical coiling proposed by others[13]. Further analyses showed that Cα atoms of the same residues between two stacked β-strands were separated at ~0.48 nm matching the expected β-strand spacing (Fig. 1d and Supplementary Figs. 3 and 5). Distances between terminal residues, two from each end, deviated to greater spacings of 0.5–0.8 nm (Fig. 1f and Supplementary Fig. 3). Thus, each helix becomes loosened and curved in its proline-rich apex to facilitate the packing of β-helices into a cylinder. In any pair of two stacked triangular rungs, distances between residues in the vertices of one rung and the opposite sides of the other rung were predictably 0.1–0.2 nm greater than the distances within individual rungs, which were estimated as 16 ± 0.8 Å (Supplementary Figs. 3 and 4). This value agrees with a dominant d-spacing of 16.5 Å in X-ray diffraction (XRD) patterns, as well as with current β-helical models[28,29], suggesting that the spacing corresponds to the diameter of the β-helix. By contrast, no spacings were observed for a control peptide, which cannot fold or assemble at the conditions (Fig. 2a). Other d-spacings obtained by XRD for SaBeH at 4.17, 4.97 and 5.85 Å (angle 2θ of 14–24°) appeared to deviate from the classical inter-strand spacing of 4.7 Å (0.47 nm) (Fig. 2a). This may be due to the varied distances including those between the terminal residues as observed by the MD simulations (Supplementary Fig. 3).

The three spacings proved to be the first orders of the other periodicities recorded indicating that the repeating units along the cylinder axis may comprise two strands (9.89 and 11.68 Å), three strands (14.9 Å) and four strands (16.5 Å). However, no second and third orders were observed for the latter value, whereas the first and second orders were apparent for 14.9 Å[30]. The X-ray counts for the spacing at 11.68 Å and its first order at 5.85 Å were low in comparison. This distinction accepts a three-strand structure as the predominant

repeating unit of the SaBeH assembly and may relate to the three β-strands of the individual triangular rung or three β-strands forming the β-sheet interface of each β-helical turn (Fig. 1e)[30,31]. With no periodicities observed at 6–9 Å region, which would otherwise indicate a generic amyloid structure[30–32], the results support the formation of a β-helical assembly, which is further supported by the analysis of the backbone dihedral angles, Φ/Ψ, in the assembly. The angles were found in a narrow range of the Ramachandran plot corresponding to β-structures, namely parallel β-sheets and turns (Fig. 2b).

Interestingly, NNV, NPP and PPI, were featured in a relatively rare region of left-handed helical conformations suggesting that these residues are involved in arranging the consecutive coiling of the rungs, which gives rise to a helical pattern[33].

The constructed three-helix cylinder proved to be stable in the MD simulations. No appreciable changes occurred in the radius of gyration and root-mean-square-deviation (RMSD) of the assembly (Supplementary Fig. 6). Some peaking within 0.5–0.75 Å was observed in the RMSD plots at ~27 ns and 115 ns for 100-ns and 200-ns simulations, respectively. The peaking was consistent with the vacillation of the terminal prolines resulting in the varied spacings in the distance maps and was deemed marginal to reflect a more global variation in the assembly (Supplementary Figs. 3 and 6). Consistent with these findings, Fourier Transform Infra-Red (FT-IR) spectra of SaBeH were characteristic of an appreciable β-structure (Fig. 2c). The second derivative spectra of the amide I region revealed a major band at 1623–1625 cm⁻¹ corresponding to inter-molecular β-strand interactions. No bands were detected at 1636 cm⁻¹ or 1689 cm⁻¹, which would have been indicative of an anti-parallel β-sheet structure[34]. A clear band observed at 1665–1668 cm⁻¹ stand for contributions from β-turn conformations in β-sheet structures, which were also confirmed by an apparent band for turns in the amide II region at ~1540–1550 cm⁻¹ (Fig. 2c)[34,35]. Intriguingly, the latter band is more common for α-helical conformations. The half-width of this band was found to be ~15 cm⁻¹ which indicates stable helices[36]. However, for α-helices the band is normally accompanied by a band at 1649–1651 cm⁻¹[37], which was not observed for SaBeH. With no unique or common FT-IR spectrum established for β-helices[38], it remains inconclusive whether the coiling of β-helical turns contributes to a specific band pattern. Therefore, the band at ~1540–1550 cm⁻¹ better fits with the combination of the bands assigned to turns. The formation of a strong β-structure was confirmed by Circular Dichroism (CD) spectroscopy, which averages all conformers present in the analyte to a principal form. The spectra recorded at micromolar concentrations exhibited spectral features reported for ideal β-helices, comprising a relatively large amplitude at ~216 nm and a maximum at ~195 nm (Fig. 2d and Supplementary Fig. 7)[39,40]. The structure remained stable upon thermal denaturation up to 80 °C, retaining the bands, while intermittent spectra recorded every 10 °C revealed an isodichroic point at 204 nm, indicating a two-state transition between β-helical and partially unfolded forms (Supplementary Fig. 7a). The thermal denaturation gave sigmoidal unfolding curves which are characteristic of a cooperatively folded structure (Supplementary Fig. 7b). The first derivative of the curves comprised a dominant transition midpoint ($T_M$) at ~65 °C (Supplementary Fig. 7c). Upon cooling the assembly was reversible as CD spectra recorded before and after the melt were nearly identical (Supplementary Fig. 7d). Moreover, CD spectra for repeated runs of heating and cooling recorded for different sample preparations at the same concentrations confirmed not only the reversibility of the fold, but also its quantitative recovery following thermal denaturation (Supplementary Fig. 7e). Thermal denaturation experiments performed under the same conditions using FT-IR spectroscopy to provide complementary evidence for the conformational properties of the fold returned a comparable $T_M$ of ~67 °C (Supplementary Fig. 7f).

As gauged by transmission electron microscopy (TEM) and atomic force microscopy (AFM), SaBeH assembled into discrete

cylindrical nanoscale structures (Fig. 2e, f and Supplementary Fig. 8a–d). Diameters of these structures were in the predicted 3 and 4 nm range and were comparable by width (TEM) and height (AFM) measurements (Supplementary Fig. 8e–g). The structures were found as anisotropic assemblies with the mean length of 62 nm (Supplementary Fig. 8g). Longer structures were also observed but did not exceed 100 nm in length, which is well below persistence lengths for amyloid, helical or collagen fibrils, whose morphologies also tend to be more heterogenous[41–43]. Dynamic Light Scattering (DLS) measurements provided further support for the microscopy data. Performed in solution for the same concentrations DLS analyses revealed an average hydrodynamic diameter of 84.8 ± 1.1 nm. Size distributions by intensity, volume and number were in good agreement with one another, despite that DLS is limited in size resolution to the factor of 3 and fails to reliably resolve narrow size distributions (Supplementary Fig. 9a). DLS measurements relate the Brownian motion of a particle to its diameter by assigning to it a hydrodynamic diameter, which for a non-spherical particle assumes the diameter of a sphere with the same translational diffusion speed as the particle. For anisotropic particles, such as the observed cylinders, changes in length will affect the size determined by DLS, but changes in diameter are harder to detect as these would have no effect on the diffusion speed. Since DLS does not capture the shape of a particle the data obtained is to complement the results of the TEM and AFM measurements which provide more accurate and direct determination of size and shape.

The average size and size distributions observed by DLS and microscopy were found to hold at concentrations above the critical aggregation concentration (CAC) of 12.5 μM (Supplementary Fig. 9b), which correlated well with high intercepts in DLS correlograms, indicating that the assembly was complete at micromolar concentrations above CAC (Supplementary Fig. 9c). Thioflavin T (ThT) fluorescence assays performed to complement the DLS data gave the same CAC (Supplementary Fig. 9d). As the fluorescence of ThT increases upon its binding to β-sheet assemblies, titrations of SaBeH into a ThT solution allowed to plot fluorescence changes versus SaBeH concentrations to confirm the CAC determined by DLS (Supplementary Fig. 9c, d). The analysis of longer structures by AFM in liquid revealed that the SaBeH assemblies exhibit a left-handed helical twist with a helical pitch of 28.5 ± 7.3 nm (Fig. 2g). This value is comparable to short pitches of small β-structured fibrils which favour highly twisted morphology, whereas pitches for amyloid-like fibrils are typically within 100–200 nm ranges[44]. Collectively, the experimental and computational analyses confirmed the conserved dimensions of the SaBeH assembly supporting the formation of a β-helical trimer.

## β-helix cylinder

The data presented provides ample evidence for either tubular or fibrillar assembly of SaBeH or both, but insufficient evidence to unambiguously distinguish between the two types of assembly. A hollow cavity is normally viewed as the main differentiator between the two. Although it is reasonable to consider that the packaging of three helices leaves a small cavity in agreement with the MD simulations (Fig. 1f) and earlier predictions for β-helices[13,20], such a cavity would not exceed 0.4 nm in diameter, which is challenging to detect. The left-handed twist in the assembly bears similarities with short β-structured and amyloid-like fibrils postulated as water-filled nanotubes[45]. Such a morphology may well be the case for SaBeH, and if proves meaningful deserves an independent study. For the purpose of this report this distinction between tubes and fibres has not been pursued. To avoid ambiguity, we refer to the SaBeH assembly as a cylindrical structure.

## β-helix carpet-like extracellular matrix

As revealed by microscopy, SaBeH assemblies did not network, branch or connect with one another at micromolar concentrations.

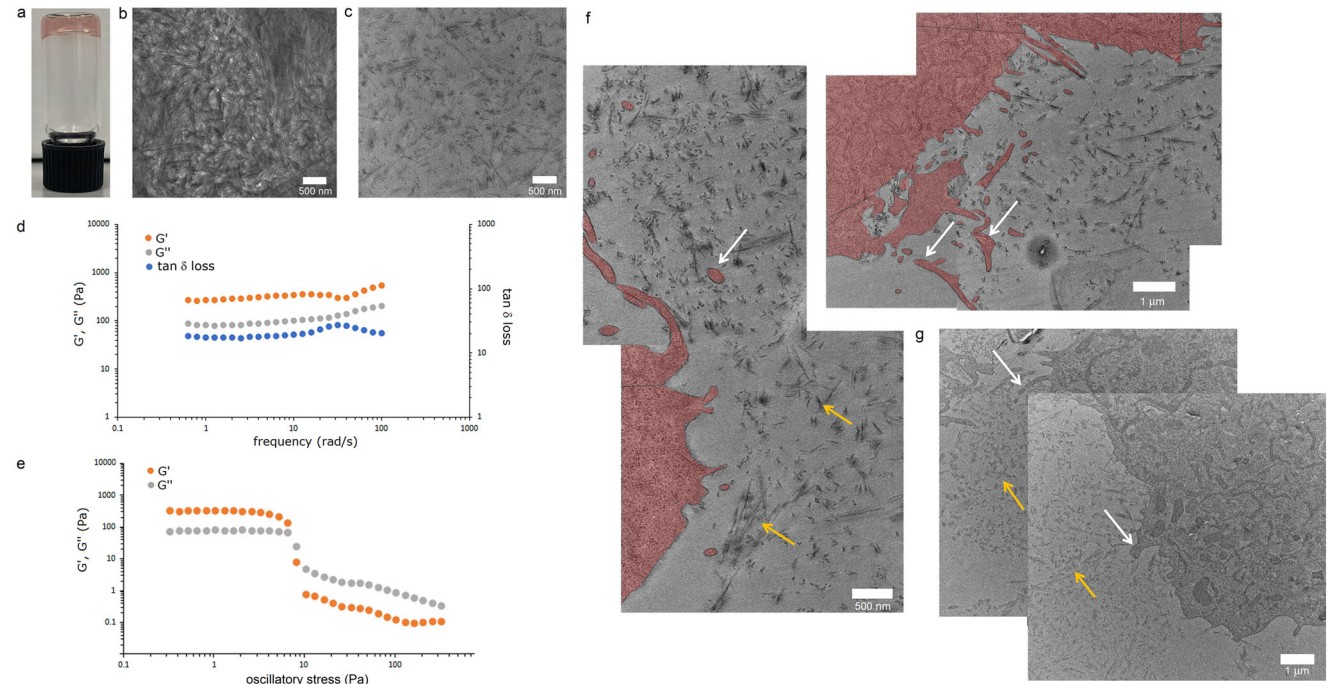

**Fig. 3 | β-helix forms hydrogel scaffold. a** A photograph of a SaBeH hydrogel (1%, w/v) set in a 5 mL vial. **b** A representative electron micrograph of the hydrogel imaged as is. **c** A representative electron micrograph of the hydrogel microtomed at 50 μm thickness. **d** Storage (G´, orange) and loss (G´´, grey) moduli of the hydrogel (1%, w/v) and the tangent of the loss angle (tan δ loss, blue) measured in the 0.5–100 rad/s frequency range. **e** Oscillatory stress sweeps of the hydrogel (1%, w/v) showing linear viscoelastic region with a cross-over point (~10 Pa) between the moduli characteristic of a gel-sol transition. **f** Representative electron micrographs of HeLa cells and **g** representative electron micrographs of MDA-MB-231 cells, grown in the hydrogel over 24 h. White arrows point to cellular lamellipodia. Gold arrows point to individual and clustered SaBeH assemblies. Cell areas (**f**) are labelled red to help visualize microtomed lamellipodia. Images in (**b**, **c**, **f**, **g**) are representative of at least 3 independent experiments.

This tendency to form discreet cylindrical nanostructures remained at higher and millimolar concentrations, typically used for gelation. Indeed, SaBeH gelled at 1%–5% (w/v) (Fig. 3a and Supplementary Fig. 10). The density of the nanostructures in gels increased dramatically resulting in a carpet-like scaffold material (Fig. 3b). No networks, void spaces or pores could be identified, suggesting that the morphology of SaBeH assemblies did not change. Clustering also increased upon gelation, which was particular evident from microtomed preparations of the assemblies (Fig. 3b, c). Since microtomes are 2D slices of 3D materials embedded in resin, the assemblies are expected to orient at varied angles with respect to the plane of the image, as observed (Fig. 3c). As the density of the nanostructures increases, the impact of their relative orientation in the microtome images becomes less apparent. When imaged as is, the material exhibited comparable density at all gelation concentrations (Supplementary Fig. 10a), while microtomed preparations of the material revealed relatively higher densities at higher gelation concentrations (2% and 5%, w/v) (Supplementary Fig. 10b). The microtome analysis confirmed the highly dense nature of SaBeH in 3D.

Complementary to this, the storage (G′) and loss (G′) moduli of 1, 2 and 5% (w/v) gels measured in the 0.1–100 rad/s frequency range were characteristic of soft gels including protein and β-structured gels (Fig. 3d and Supplementary Fig. 10c)[46–48]. No cross-over between G′ and G″ was observed for the gels suggesting that the gel remained largely elastic over the frequency range and concentrations used[49]. The elasticity of the gels can be attributed to its dense fibrous structure dominating its mechanical properties[46]. At high frequencies (>30 rad/s), tan δ loss decreased indicating increased elasticity due to the dynamic stiffening of the gel microstructure (Fig. 3d and Supplementary Fig. 10c)[47]. However, contrary to linearly elastic hydrogels, the SaBeH gel exhibited a more complex behaviour. When a shear force was applied, the gel responded as a non-Newtonian fluid demonstrating a strong shear-thinning behaviour manifesting in the gradual loss of viscosity for all gel concentrations as the shear rate increased (Supplementary Fig. 10d). This effect is consistent with the anisotropic assemblies of SaBeH aligning along the direction of the shear force, resulting in the increased free space between them and reduced viscosity of the gel. This is also intriguing as for SaBeH G′ was greater than G″ (Fig. 3d and Supplementary Fig. 10c), suggesting a well-developed fibrous network in the gels, whereas the two moduli remained relatively low (<1 kPa) in the frequency range used, indicating that the gels had a low density of physical cross-linking in its microstructure. The same behaviour held true for 5% gels, for which marginally increased G′ (≥1kPa) and tan δ loss decreasing at higher frequencies (>60 rad/s) suggest increased fibrous density, which was confirmed in the microtomed preparations (Supplementary Fig. 10b, c). Furthermore, oscillation stress sweeps performed to probe the microstructure of the gel revealed that the linear viscoelastic region of the gel was shortened at ~10 Pa—a cross-over point at which G′ and G″ had the same values (Fig. 3e). As expected for a gel form, G′ was greater than G″ at lower values, but lower than G″ in the stress region (>10 Pa) where transition from the gel into a liquid form occurred. Consistent with the classical shear-thinning behaviour found for the gels (Supplementary Fig. 10d), which is also observed upon increasing shear in peptide, polysaccharide, and polymer hydrogels exhibiting dynamic fibrous, cross-linked, or reversible covalent networks[50–53], this finding indicates that the dense fibrous microstructure of SaBeH supports the flow of SaBeH gels when stress is applied. Native extracellular matrices (ECMs) such as collagens and fibrins, whose viscoelastic properties are exploited to promote cell growth, are formed by micron-sized filaments capable of encapsulating cells in 3D[49].

These exemplar materials comprise complex, highly porous networks of protein filaments exhibiting high persistence lengths. Such properties are believed to impact on the ability of cells to proliferate and migrate[49]. The SaBeH scaffold offers contrasting properties— highly dense, carpet-like, non-porous and non-networking deposits of extremely short nanoscale cylinders. Since the impact of such properties on cell responses has yet to be shown for any material, SaBeH gels were assessed in cell cultures. Native collagen type I ECM (NCTI) was used for comparison. As observed in the microtomes of cell-laden gels, the SaBeH assemblies retained their morphology as well as carpet-like density (Fig. 3f, g), which were similar to those of the gels without cells (Fig. 3c). An appreciable formation of cellular lamellipodia was observed indicating that cells adopt an adhesion-based migration mode in 3D and can proliferate.

Indeed, both human immortalized cells (cervical cancer, HeLa) and human primary cells (human dermal fibroblasts, HDFs) proliferated steadily over time (Fig. 4a, b and Supplementary Fig. 11a). PrestoBlue™ cell-proliferation assays revealed comparatively low growth rates in SaBeH than in NCTI gels (Fig. 4c). The assays rely on the chemical redox reaction of resazurin converting into resorufin at the reducing environment of metabolically active cells (pH >6). The isoelectric point (pI) of SaBeH is 10, whereas that of NCTI is 7-8. Unlike NCTI, which is an amphoteric molecule, SaBeH carries a strong positive charge in the extra- and intracellular environments (pH 7–7.8). Since cationic peptides inhibit the reduction of resazurin, and more significantly at higher concentrations[54], proliferation rates were monitored by fluorescence microscopy. Based on a direct cell count differences in cell proliferation between the two gels were apparent (Fig. 4d). In general, cell growth rates did not change significantly over time for either of the two. However, HeLa cells and HDFs tended to proliferate more in NCTI and SaBeH, respectively. The tendency was confirmed by MTT™ cell viability assays, which gave higher numbers of metabolically active cells for HDFs when compared to HeLa cells grown in the SaBeH gels (Supplementary Fig. 11a). Cell viability measured by MTT™ assays was comparable to that measured by PresoBlue™, while the results did not depend on the concentration of the tetrazolium (MTT) dye, suggesting that SaBeH had no effect on its conversion (Supplementary Fig. 11b). Further evidence for SaBeH supporting functional 3D cell culture was sought in Lactate dehydrogenase (LDH) release assays. Unlike metabolic assays, these assays measure the levels of an endogenous cytosolic enzyme, i.e., LDH, which is released upon damage to the cell membrane and therefore provide a straightforward probe for cytotoxicity. The assays showed that SaBeH outperformed NCTI in providing non-toxic environment for HDFs, which was in good agreement with the tendency for SaBeH to promote the growth of the cells over HeLa cells established by the metabolic assays (Supplementary Fig. 11c).

To this end, all the three assays used rely on enzymatic conversions of reporter molecules or metabolites providing quantitative insights into the metabolic responses of the cells to their extracellular environments, i.e., scaffolds. Strictly speaking, such responses are secondary to proliferation rates defined at the genetic level in particular during DNA synthesis. With this in mind, proliferation assays using (5-ethynyl-2′-deoxyridine) (EdU) were performed. EdU is incorporated into newly synthesized DNA in dividing cells thereby providing a direct measure of cell proliferation. Monitored by flow cytometry, the assays confirmed consistently high proliferation rates for both cell types over 72 h of incubation in SaBeH. The results strongly support that SaBeH did not affect the DNA replication phase, which can serve as a checkpoint control of proliferation, at any point of cell division (Supplementary Fig. 11d, e). No appreciable difference in cell proliferation could be ascertained between HDF and HeLa by the assay either. This is in contrast to that observed by metabolic assays, indicating that SaBeH had no impact on cell division. Preferences observed by metabolic assays towards supporting the growth of HDFs more than

HeLa cells are likely to result from the differences in cell membranes of the two cell types. Characteristic of cancer cells, HeLa membranes are more anionic than HDF membranes and are more prone to binding to cationic SaBeH, which may impact on their proliferation.

A metabolic assay based on long-term dye retention, such as CellTrace Violet, can provide complementary evidence (Supplementary Fig. 11f, g). Monitored by flow cytometry this assay is designed to relate cell divisions, with subsequent divisions giving larger cell numbers, with fluorescence intensity, which decreases with every division. Higher fluorescence intensities were recorded for HDFs at any given time point measured over 168 h (Supplementary Fig. 11f). A general tendency of decreasing intensity was observed for both cell types. However, intensities and cell numbers for HeLa were comparable for 24 and 72 h.

This is intriguing as sharp increases in cell numbers were observed for HeLa at each subsequent time point, while median fluorescence for 168 h was found to be nearly threefold lower than that for 72 h (Supplementary Fig. 11f, g). With greater cell counts recorded for HeLa in 2D without matrices (Supplementary Fig. 11g), this may suggest a comparatively lagged proliferation of HeLa cells in SaBeH resulting from the need for cells to adjust in 3D, e.g., through focal adhesions, spreading and clustering, the patterns of which are defined by the scaffolds, and which was indeed observed (Fig. 4). Similarly, drastic increases in cell numbers were observed for HDFs between 24 and 72 h which may also suggest cell adjustment in SaBeH (Supplementary Fig. 11g). Further, cell numbers at 168 h for HDFs and HeLa cells were comparable in SaBeH, whilst differences in median fluorescence between 72 and 168 h for HDFs were found to fit an ideal scenario involving several cell divisions each contributing to decreasing fluorescence (Supplementary Fig. 11f, g).

Taken together the results of cell proliferation and viability assays indicate that SaBeH supports cell development at all genetic and metabolic levels without cell toxicity or inhibition but is likely to introduce spatial constraints that require cells to adjust to, which may manifest in differences in cell morphologies.

In accord with this, the morphologies of individual cells proved to be notably distinctive in SaBeH when compared to those observed in NCTI (Fig. 4a, b). Both HeLa and HDFs appeared spread and multipolar in SaBeH (Fig. 4a, b and Supplementary Fig. 12a, b).

By contrast, the round morphology of HeLa cells was more appreciable in NCTI, while HDFs in NCTI maintained either plump or bipolar, unidirectional spindle shapes. Contrary to 2D scenarios, in which ECMs provide stiff substrates restricting cell attachment to one plane, whilst supporting profound cell spreading, 3D matrices are soft substrates in which cells must reach out to constituent fibres to build a network of focal adhesions sufficient for growth[55]. This prompts the formation of lamellipodial projections.

The SaBeH gels provide cells with a dense, carpet-like matrix of short cylinders. Such a substrate offers abundant focal adhesions, easing the requirement for long lamellipodia, thus giving rise to multipolar cell morphologies with shorter projections (Fig. 4e). This is in contrast to collagen matrices which present long-range fibre networks promoting long, but fewer lamellipodia and consequently limiting cell polarity to thin, bipolar morphologies (Fig. 4f)[55–57]. Thus, the cell responses observed to NCTI were consistent with those of collagens as soft 3D substrates[57].

The stellate shapes of HDFs observed in SaBeH are common for HDFs capable of building focal adhesion networks, which is also characteristic of cell responses to viscoelastic ECMs promoting mechanical cell responses more effectively than elastic scaffolds[49]. This is intriguing. Unlike NCTI, SaBeH does not contain known cell adhesion motifs, binding sites or receptor ligands. Therefore, in the absence of established biology, the basis of cell responses to the SaBeH scaffold must be due to its physical or morphological properties. Since SaBeH shares physical properties with other viscoelastic

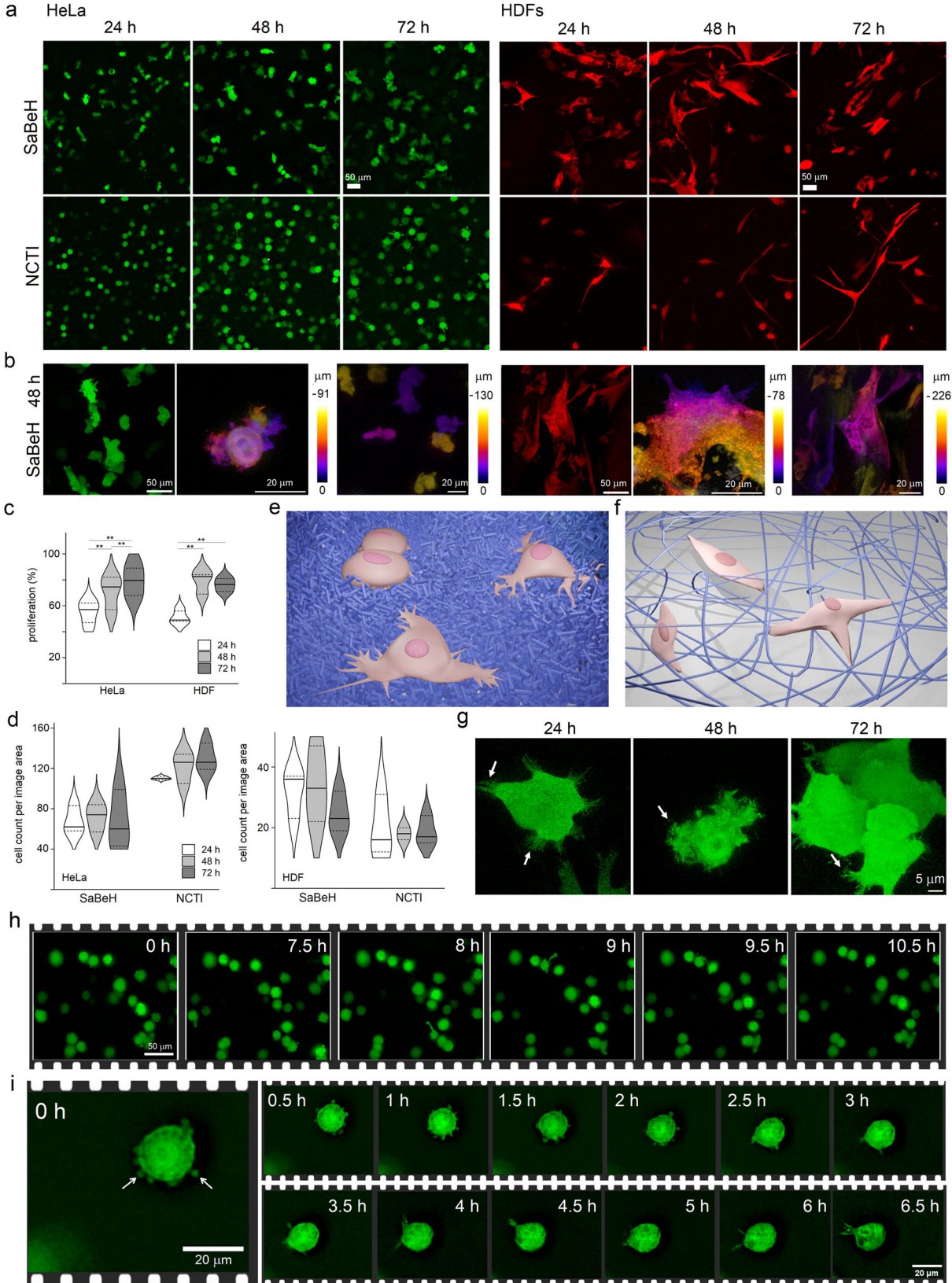

cell-supporting hydrogels based on native ECM proteins such as NCTI, synthetic peptides, polysaccharides, and polymers[46–53], the observed cell responses cannot be merely due to the bulk properties of the scaffold such as stiffness or viscoelasticity. Differences in cell behaviour are manifested in clear changes in the morphologies of the same cells seeded in NCTI and SaBeH and hence derive from cell

interactions with the microstructure of the scaffold, i.e., the dense, carpet-like matrix of SaBeH versus long-range networks of NCTI (Fig. 4e, f).

Further, like HDFs, HeLa cells maintained multipolar and spread shapes in the SaBeH scaffold. The cells profusely expressed filopodia on lamellipodia and favored cell-to-cell interactions leading to the

**Fig. 4 | β-helix supports 3D cell culture. a** Low and **b** high magnification fluorescence micrographs of HeLa cells and HDFs, expressing green fluorescent protein (GFP, green) and TurboFP602 (red), respectively. The cells grown in SaBeH and NCTI gels were imaged at fixed time points. **b** Cells located at different depths from the plane of the image are labeled with false colours matching corresponding rainbow scales indicating depth profiles. **c** Cell proliferation determined by PrestoBlue™ assays at 24 h (white), 48 h (light grey), 72 h (dark grey) after subtracting background readouts (bare gels). Total number of metabolically viable cells grown in NCTI was taken as 100%. Data are presented as average percentages of metabolically viable cells with standard deviations for six independent biological replicates ($n = 6$): HeLa (24 h: 55 ± 8.9; 48 h: 70.5 ± 12.5; 72 h: 79.3 ± 12.3); HDF (24 h: 51 ± 5.3; 48 h: 79 ± 8.4; 72 h: 76.3 ± 5.2). Solid horizontal lines denote median, and upper and lower edges correspond to 75th and 25th percentiles, respectively. Statistical analyses were done using the analysis of variance (ANOVA) followed by Bonferroni and unpaired, two-sided $t$ tests. Significant differences are represented with ** for $p < 0.01$. HeLa: $p = 0.00026$, $p = 0.00003$, $p = 0.0069$; and HDF: $p = 0.00025$, $p = 1.86 \times 10^{-5}$—significantly higher numbers of metabolically active cells were observed at 72 h than at 24 and 48 h, and significantly higher numbers at 48 h than at 24 h for HeLa cells; and at 72 and 48 h than at 24 h for HDFs. **d** Cell count of metabolically active cells per image area determined at 24 h (white), 48 h (light grey), 72 h (dark grey) by automated segmentation using StarDist (https://github.com/stardist/stardist).

Data are presented as average cell counts per image area with standard deviations for three independent biological replicates ($n = 3$): HDF/SaBeH (24 h: 32 ± 7.8; 48 h: 34 ± 12.5; 72 h: 24.6 ± 6.6); HDF/NCTI (24 h: 19.6 ± 10; 48 h: 18 ± 2.8; 72 h: 18.6 ± 4.7); HeLa/SaBeH (24 h: 67.6 ± 13.4; 48 h: 71.6 ± 13.6; 72 h: 67.3 ± 28.7); HeLa/NCTI (24 h: 110.3 ± 1.5; 48 h: 121.6 ± 15; 72 h: 130 ± 13.4). Solid horizontal lines denote median, and upper and lower edges correspond to 75th and 25th percentiles, respectively. **e** Schematic representation of multipolar cell morphologies with shorter projections (pink) on the SaBeH carpet-like scaffold (blue). **f** Schematic representation of cells (pink) exhibiting limited polarity on long-range networks of collagen fibres (blue). **e, f** Schematics were created using Blender, a free and open-source 3D creation suite. Blender is distributed under the GNU General Public License (GPL), which grants users the freedom to use, modify, and distribute the software for any purpose, including commercially and for education. Scaffolds and cells are shown in blue and pink, respectively. **g** High-mag fluorescence micrographs of HeLa cells (green) grown in SaBeH gels taken at fixed time points. White arrows indicate filopodia. **h** Time-lapse micrographs of HeLa cells (green) grown in SaBeH gels over the first 10 h. **i** Time-lapse micrographs of a representative HeLa cell (green) in SaBeH gels showing continuous re-arrangements of filopodia (white arrows) over first several hours culminating in a bleb formation (6.5 h). Images in (**g, h, i**) are representative of at least 3 independent biological replicates. Source data are provided as a Source Data file.

formation of spheroids, which increased in size over time and were comparatively larger than those formed in NCTI (Fig. 4g and Supplementary Fig. 12c). As observed by light sheet fluorescence microscopy, which allows monitoring the position of individual cells in 3D, the cells were able to adhere and migrate in the gels (Fig. 4h and Supplementary Fig. 12d).

Although changes in the location of individual cells in 3D at different depths are not deemed straightforward to determine unambiguously and did not seem to be significant for HeLa cells in SaBeH over several hours, the migration of individual cells was evident (Fig. 4h and Supplementary Fig. 12d). Individual cell tracks in the same field of view gave travel trajectories covering total distances of up to 80 μm over the first 5–10 h (Supplementary Fig. 13a–c and Supplementary Movies 1 and 2). Such distances and migration rates, even without adding up travel trajectories in 3D at different depths, were comparable to those of individual and collective cell migrations in 2D and 3D reported by others[58,59]. Cell migration is often heterogeneous and exhibits diffusion patterns that are anomalous to the Brownian motion (Supplementary Fig. 13c)[60,61]. Such patterns contain both directionality, which is expressed in linear migration trajectories, and tumbling, which is expressed in random trajectories[62–65]. Cell tracks in SaBeH gels measured individually gave broad distributions of displacements per cell (Supplementary Fig. 13b), but together coalesced into anisotropic patterns, which were more apparent for longer timescales (Supplementary Fig. 13d). The ensemble averaged analysis of these trajectories, such as the mean-squared displacement (MSD), which averages cell positions and the ensemble of their migration trajectories, confirmed an anomalous cell diffusion in the SaBeH gels (Supplementary Fig. 13e)[60–65]. The MSD curves appeared to be nonlinear and consistent with superdiffusivity patterns suggesting a degree of persistence in migration trajectories[62–65]. Indeed, this was observed. Persistence, expressed as a ratio of a net displacement versus a total distance travelled by a cell[65], was characteristic of a directional migration, which appeared to increase over time (Supplementary Fig. 13f). This tendency was reversed to that of velocity, which decreased over time (Supplementary Fig. 13g), indicating that cell migration became more directional, but slower.

The observed migration dynamics were comparable to those described by others in 2D and 3D for different cell types including actively migrating cells such as T-cells, astrocytes, leukocytes and dendritic cells[60–65]. Individually net distances travelled by cells in the SaBeH gels in the first hours were shorter suggesting some preference for comparatively more confined movements. Since persistence is

determined by the ability of a cell to build a network of focal adhesions in the extracellular matrix, such a behaviour is reflective of the stellate polarity of cells in SaBeH gels. Multiple filopodia and lamellipodia lower the membrane tension of a cell, reducing its polarity and the directionality of movement[66]. When the membrane tension increases, multiple protrusions are suppressed in favour of a main polarised protrusion which sets up a streamlined movement[67]. This scenario can be observed in individual cells grown in SaBeH gels: multiple filopodia form and continuously re-arrange until a dominating protrusion emerges (Fig. 4i and Supplementary Fig. 14a). Profound membrane blebbing can then ensue supporting an amoeba-like behaviour that is typical of cancer cells (Supplementary Fig. 14b)[68], which employ blebbing for locomotion within their extracellular environment and extrusion from epithelia prior to metastasis[69]. Blebbing allows cells to release intracellular pressure by forming polarised protrusions—lobopodia at a leading end of the cell and uropodia at the rear end[69]. The cell mass moves towards lobopodia, whilst adhesive bonds between uropodia and the ECM are broken thereby promoting de-adhesion. Because lobopodia protrude faster than they can adhere to the ECM, the cell can move forward. Primary cells, such as HDFs, use similar mechanisms for migration by switching between polarised protrusions of low and high pressure which leads to their bi- or multi-polar shapes, as was observed in SaBeH gells[70]. Taken together the results suggest an interplay between the stellate morphology of the cells, which maintain multiple protrusions in SaBeH, and the evolution of the protrusions into blebbing, which determines the anomalous diffusion of the cells in SaBeH gels via low-adhesion migration modes. Thus, the results support the view of the current models that the dynamics of cellular migration is executed at complexity levels that is well beyond the individual components of cell migration[60–67].

Indeed, critical manifestations of cell migration are complex and include the leading-edge extensions, i.e., abundant lamellipodia and filopodia observed in SaBeH; cell adhesion to the matrix, i.e., distinctive multipolar and spread morphologies observed for cells in SaBeH; cell-cell interactions, i.e., cell clustering leading to spheroid formation; and release from contact sites, i.e., cell migration at distances over tens of microns in 2D alone.

Since cell migration depends on a number of factors including a proper interpretation of external cues by migrating cells and since there is no known biology incorporated into SaBeH, the cues the cells recognise are physical, but not necessarily normal for cells, which reflects in morphological changes of the cells and their lagged proliferation resulting from the need to adjust in the SaBeH environment.

In this vein, SaBeH reduces the membrane tension of adhering cells while promoting their growth but lacks specific biological cues that could streamline the trajectories of cell migration[60,65]. Because SaBeH forms a dense, carpet-like matrix of chemically identical nanoscale cylinders, which cells profusely attach to, any cues that can straighten the directionality of travel should be in low abundance. This creates the potential to tune the design. By incorporating site-specific biological signals or binding sites in the SaBeH matrix as a whole it should be possible to tailor the cell migration dynamics in favour of preferentially unidirectional or confined modes. With no known and pre-defined biology in the reported SaBeH design, the observed cell adhesion and migration may not be specified by biological cues. Unless SaBeH incorporates effective, yet unknown signals, cell migration in SaBeH gels is more likely to derive from the physical properties of the scaffold whose carpet-like microstructure provides virtually an unlimited space for focal adhesions (Fig. 4e, f). Given that the scaffold has no persistent networks, the focal adhesions formed are short-lived but readily rearrange, which necessitates their abundant formation manifesting in multipolar cell morphologies, promoting amoeboid modes of cell migration[67–70].

The cylinders are also cationic, which contrasts with negatively charged collagen fibrils. This distinction prompts a conjecture that the cylinders may be able to translocate across cell membranes, complex with nucleic acids and deliver them into the cell. The positive charge of SaBeH is posed to mediate interactions via heparin with primary cell membranes, which are otherwise zwitterionic, and with immortalised cells, which carry a negative charge due to the elevated levels of anionic glycoproteins on their surfaces. Since the cylinders do not match the cell length scales, membrane binding can develop into cell penetration via an endosomal proton sponge mechanism, pore formation or direct translocation. Polycations, protein transduction domains, cell penetrating peptides, and capsids-like protein assemblies of similar sizes promote the intracellular delivery of nucleic acids[71–73].

### β-helix promotes gene delivery in 3D

To probe this, SaBeH was co-assembled with a small interference RNA (siRNA) interfering with the expression of GFP. With the ζ-potential of the resulting assemblies supporting encapsulation (27.4 ± 0.18), high levels of intracellular uptake were observed at low charge ratios (CRs) of SaBeH to siRNA at micromolar concentrations without toxicity (Fig. 5a, b and Supplementary Fig. 15a, b). Similar uptake levels without toxicity were observed for SaBeH co-assembled with a plasmid DNA (pDNA) encoding for a fluorescence protein iLOV at the same concentrations and CRs. Live cell imaging using confocal fluorescence microscopy revealed abundant florescent patterns of fluorescently labelled siRNA and pDNA within the first hours of transfection. The patterns appeared as clusters concentrating towards the perinuclear regions of the transfected cells. Transfections mediated by lipofectamine (LF), used for comparison, gave characteristic particulate patterns showing no distribution preferences in the cytoplasm (Fig. 5a). Complementary to these observations, the cell uptake as a function of fluorescence was quantified using flow cytometry (Supplementary Fig. 15a, b). Both median and mean fluorescence for the labelled siRNA and pDNA were found to be significantly higher than those for LF (Fig. 5b). These results indicate effective intracellular gene delivery, though gene knockdown (siRNA) and expression (iLOV) were not significant, showing positive responses only in <20% of all transfected cells. Regardless, however, this ability of SaBeH to deliver nucleic acids opens up an opportunity for a biomaterial with a dual function[74]. Whilst supporting the growth and proliferation of cells in 3D, such a material can also deliver genetic material into the same cells creating a means of combining the impacts of mechanotransduction and genetic manipulation on cell development by the same extracellular niche without the need of using gene delivery vectors (Fig. 5c). To gain a better insight into this possibility, the 3D cell culture experiments were repeated using the SaBeH gels assembled with the pDNA encoding for iLOV. As gauged by fluorescence microscopy, apparent iLOV fluorescence was observed in <30% of the transfected cells, consistent with gene expression levels in 2D (Fig. 5d). Taken together these results demonstrate that SaBeH can indeed support a synergistic function of promoting cell growth and intracellular gene transfer in 3D.

The exact mechanism whereby SaBeH promotes intracellular delivery remains to be elucidated and merits a separate study. Nonetheless, one can speculate that with SaBeH lacking known specialist biology, such a mechanism is likely to be similar to those employed by cationic nanoscale encapsulators and condensation reagents. Like the reagents of this type, SaBeH is a nanoscale, amphipathic and cationic agent, which packages and delivers genetic material into the cell without cytotoxicity (Fig. 5 and Supplementary Fig. 15). To gain a better mechanistic insight into the intracellular delivery of genetic material by SaBeH, cells before transfections were treated with cell uptake inhibitors, and the results obtained were compared with transfections without the treatments and transfections performed using a cationic encapsulator (LF) and a cationic condensation reagent (HaL°). As gauged by flow cytometry, preincubations with $NH_4Cl$, which blocks endosomal acidification and consequently egress, caused 20–30% decreases in fluorescence when compared to that of untreated cells. With similar decreases observed for LF and no decreases observed for HaL°, with both reagents known to undergo endocytosis, the results indicate that SaBeH may remain entrapped in the endosomes, or its endosomal escape is independent of acidification. However, pre-treatments with antibiotic bafilomycin A1 (Baf A1), which arrests both endosomal uptake and acidification, had no effect on SaBeH-promoted transfections (Supplementary Fig. 15c). By contrast, reduced uptake was observed by the other two reagents which is indicative of inhibited both endosomal uptake and escape. The same levels of cell uptake by SaBeH with and without the treatment of Baf A1 may suggest a non-endosomatic route of cell entry. However, the effect by Baf A1 relies on the inhibition of ATPase proton pumps regulating sub-membranous pH, which relates to energy dependence, as well as the formation of macropinosomes. To test if the cell entry was energy-dependent, the cells were pre-treated with inhibitors of ATPases−sodium azide and deoxyglucose. Similar levels of reduction in the uptake were observed for SaBeH-transfected cells pretreated with $NH_4Cl$, which is consistent with an energy-dependent uptake characteristic of peptide transfections reagents, such as HaL°, and in contrast to energy-independent pathways common for liposomal reagents[75], such as LF (Supplementary Fig. 15c). Collectively, the results indicate that SaBeH exhibits an energy-dependent and acidification-independent cell uptake with no dependence or limited capacity for endosomal egress. The latter can also be impacted by the efficacy with which SaBeH encapsulates nucleic acids. Sybr™ Gold intercalation assays, which are used to detect free nucleic acids and hence quantify their encapsulated forms, gave nearly quantitative gene encapsulation (up to 94%) at low CRs (Supplementary Fig. 15d). DNA retardation gels provided complementary evidence for this. No electrophoretic mobility was observed for SaBeH co-assembled with pDNA at the CRs whereat the encapsulation was essentially complete. This was in contrast to pDNA used bare and mixed with the non-assembly control peptide even at higher CR4, which exhibited the electrophoretic mobility expected for its size (Supplementary Fig. 15e). In addition, these preparations did not survive treatments with DNase I whereas the enzyme had no apparent effect on pDNA co-assembled with SaBeH confirming that pDNA was protected by encapsulation (Supplementary Fig. 15e). Consistent with their positive ζ-

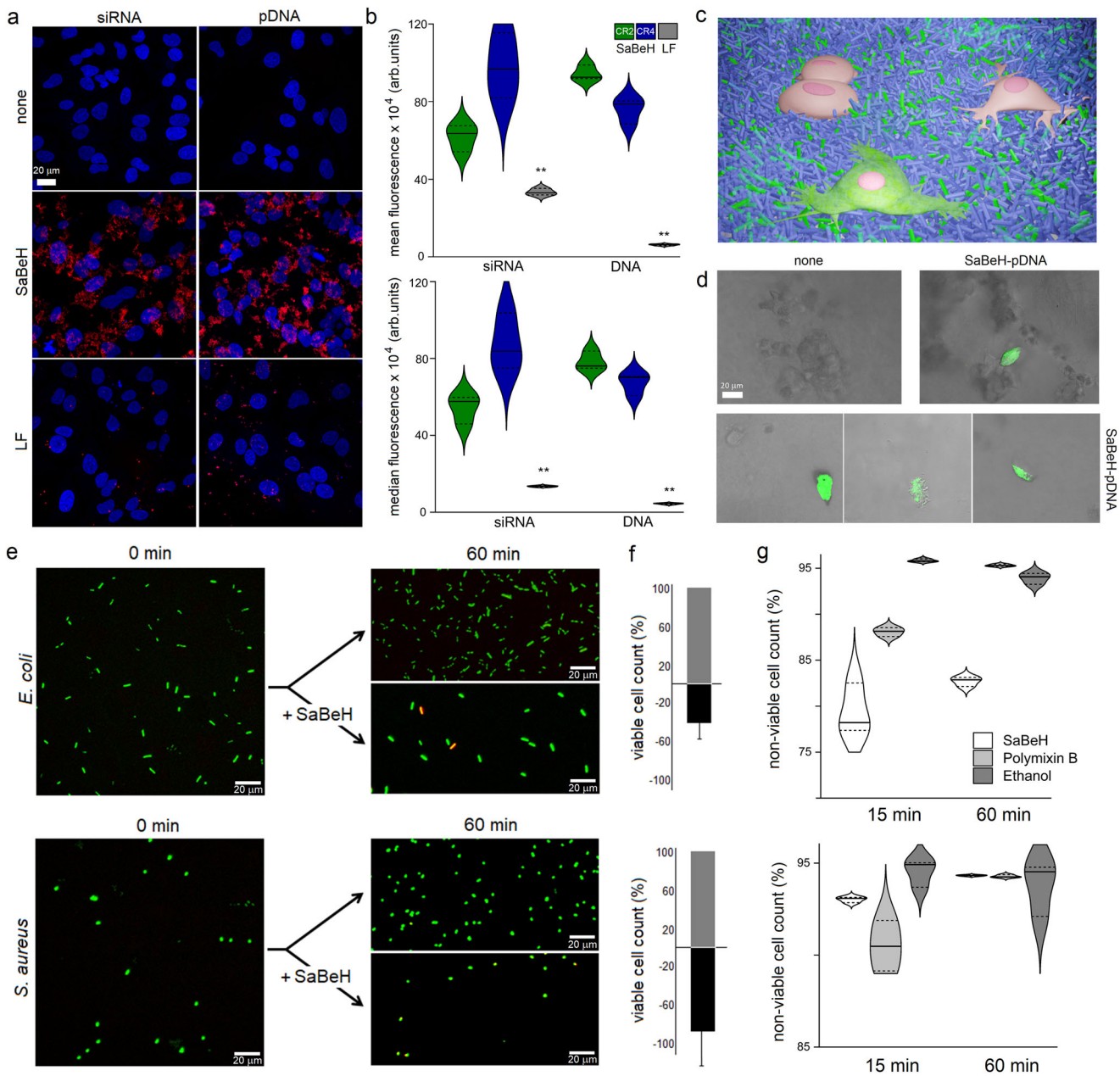

potential values, the SaBeH assemblies remained cationic under the conditions used, and thus delivering genetic cargo into the cell as nanoscale, cationic agents.

## β-helix inhibits bacterial growth

Notwithstanding differences in mechanisms, the cationic and amphipathic nature of the SaBeH assembly is likely to mediate its translocation across cell membranes. This property alone allows to discriminate between prokaryotic and eukaryotic cells. Therefore, the impact of the assembly was assessed against bacteria. Metabolic assays, such as PrestoBlue™, revealed that the inhibition was concentration dependent, with the bacterial growth gradually decreasing with increasing peptide concentrations (Supplementary Fig. 16a). However, considering potential effects of cationic peptides on metabolic conversions, even though these may be less pronounced in bacterial culture than in mammalian culture, these results were deemed inconclusive. Therefore, we sought complementary evidence. When monitored by live-cell imaging, bacterial growth in the presence of SaBeH was found to be inhibited within the first hour of incubation

(Fig. 5e, f). Flow cytometry measurements performed under the same conditions confirmed the results (Fig. 5g and Supplementary Fig. 16b). SaBeH proved to inhibit bacterial growth, which was consistent with the efficacy of membrane-active antibiotics, such as polymyxin B used as a positive control (Fig. 5g).

As reported elsewhere[25,76,77], peptides and proteins with no apparent antimicrobial activity can self-assemble into matrices capable of entrapping bacteria and resisting bacterial colonisation. Consistent with these reports, Minimal Inhibitory Concentrations for SaBeH measured against a panel of Gram-positive and Gram-negative pathogens were high (>50 μM), suggesting that inhibitory effects caused by SaBeH occur upon its assembly. Specifically, as shown in Fig. 1f, the SaBeH assembly exhibits a contiguous cationic seam, a lysine ladder, which provides a more substantial cationic surface than an individual molecule to promote strong binding to bacterial cells. Upon binding SaBeH cylinders can then incorporate into phospholipid bilayers where their hydrophobic residues (V, I and W) re-arrange to engage with lipid aliphatic chains. With the charge interactions maintained between lysine residues and phospholipid headgroups, these

**Fig. 5 | β-helix promotes intracellular gene delivery and inhibits bacterial growth. a** Fluorescence micrographs of HeLa cells transfected with siRNA and pDNA (200 ng) labelled with Alexa 647 and Cy5 (red), respectively, using SaBeH (CR4 for siRNA and CR2 for pDNA). Lipofectamine (LF) is used for comparison. Hoechst 33342 (blue) was used to stain nuclei. **b** Cell uptake for siRNA (blue) and pDNA (orange) determined by flow cytometry at 3 h post-transfection with SaBeH at CR2 (green) and CR4 (blue) and LF (grey). Data are presented as the (upper) and median (lower) fluorescence with standard deviations for three independent biological replicates ($n = 3$): siRNA/mean $\times 10^4$ (CR2: 61.7 ± 6.8; CR4: 97.9 ± 16.9; LF: 33.3 ± 1.7); DNA/mean $\times 10^4$ (CR2: 94.4 ± 3.9; CR4: 75.8 ± 6.3; LF: 6.1 ± 0.4); siRNA/median $\times 10^4$ (CR2: 54.4 ± 7.4; CR4: 87.4 ± 14.5; LF: 13.5 ± 0.3); DNA/median $\times 10^4$ (CR2: 78.2 ± 4.8; CR4: 67.2 ± 5.6; LF: 4.4 ± 0.3). Solid horizontal lines denote median, and upper and lower edges correspond to 75th and 25th percentiles, respectively. According to one-sided ANOVA tests, the counts of transfected HeLa cells for siRNA and pDNA when complexed with LF were significantly lower than those complexed with SaBeH. Statistically significant differences are represented with ** for $p < 0.01$: mean fluorescence siRNA ($p = 0.0009$), mean fluorescence DNA ($p = 6.37 \times 10^{-7}$), median fluorescence siRNA ($p = 0.00023$), median fluorescence DNA ($p = 1.5 \times 10^{-6}$). **c** Schematic of 3D SaBeH scaffold (blue cylinders) and when complexed with pDNA (green cylinders) showing cells without (pink) and with (green) iLOV expression. The Schematic was created using Blender, a free and open-source 3D creation suite. Blender is distributed under the GNU General Public License (GPL), which grants users the freedom to use, modify, and distribute the software for any purpose, including commercially and for education. **d** Fluorescence micrographs of HeLa cells grown in SaBeH gels without (none) and with the unlabelled, iLOV-encoding pDNA (SaBeH-pDNA). The micrographs highlight cells with significant iLOV fluorescence (green) and are representative of at least 3 independent biological replicates. **e** Fluorescence micrographs of *Escherichia coli* and *Staphylococcus aureus* cells grown over 60 min without (none) and with SaBeH at 100 μM. **f** Viable cell counts without (grey) and with (black) treatment with SaBeH (100 μM) over 60 min given in percentage. Negative values denote total counts of dead bacterial cells obtained by subtracting counts of viable bacteria grown without SaBeH (100%) from counts of viable bacteria grown with SaBeH. Data are presented as the percentage of the total counts of dead bacteria with standard deviations for three independent biological replicates ($n = 3$): SaBeH treated *E. coli* (−40 ± 19) and *S. aureus* (−87 ± 43). **g** Quantification of non-viable *E. coli* and *S. aureus* cells treated over 15 and 60 min with SaBeH (white), polymyxin B (light grey) and 70% aq. (v/v) ethanol (dark grey) after subtracting corresponding counts obtained for untreated cells, measured using flow cytometry. Data are presented as the percentages of non-viable bacteria cells in the total number of the cells with standard deviations for three independent biological replicates ($n = 3$): SaBeH treated *E. coli* (15 min: 79.3 ± 2.7; 60 min: 82.7 ± 0.5) and *S. aureus* (15 min: 93 ± 0.15; 60 min: 94.3 ± 0.03); polymyxin B treated *E. coli* (15 min: 88 ± 0.49; 60 min: 95.3 ± 0.12) and *S. aureus* (15 min: 90.5 ± 1.3; 60 min: 94.2 ± 0.07); 70% aq. (v/v) ethanol treated *E. coli* (15 min: 95.8 ± 0.16; 60 min: 93.9 ± 0.6) and *S. aureus* (15 min: 94.5 ± 0.74; 60 min: 93.8 ± 1.4). Solid horizontal lines denote median, and upper and lower edges correspond to 75th and 25th percentiles, respectively. Source data are provided as a Source Data file.

hydrophobic contacts disrupt bacterial membranes and inhibit bacterial growth in a manner analogous to that shown for other fibrillar matrices[25,76,77].

## Discussion

Protein folding motifs underpin the complexity of mechanistic biology. Many motifs have conserved roles, e.g., collagen triple helix. Others appear to be more versatile, e.g., α-helical coiled coils. However, structural features and functions, which are not characteristic of native proteins, are accessible by design. Arguably, any motif can be re-designed and re-purposed for functions that are deemed atypical to it. Proteins that incorporate β-helices support various functions without belonging to a particular class of globular or fibrous proteins, which creates a precedent for design. Surprisingly, however, the information available on β-helices is scarce. Furthermore, like β-sheets or α-helices, β-helices can assemble into higher-order structures, e.g., prions, but unlike β-sheets or α-helices, which can also form as a result of self-assembly, a self-assembled β-helix had yet to be shown. This study fills this gap by designing such a structure. The structure introduces several exploitable features.

First, we have demonstrated that an elementary unit consisting of three consecutive hexapeptide repeats forming three β-strands separated by turns, which constitute one β-helical coil, is sufficient to fold into a β-helix via self-assembly. The unit forms an equilateral triangle whose terminal residues non-covalently abut to accommodate the axial rise of 0.47 nm, which results in extended β-helices formed in a non-covalent fashion and trimerized into a discreet cylinder with a classical left-handed twist. The cylinder exhibits characteristic ladders of stacked residues—the property we capitalised upon to endow the assembly with appreciable functions such as differential membrane binding, cell and gene encapsulation and cell penetration. Second, with increasing concentrations of the unit the density of β-helix cylinders per unit volume increases without apparent changes to their uniform, discreet morphology. At 1–5% concentrations (w/v) the cylinders furnish highly dense, carpet-like, non-porous and non-networking scaffolds that gel. The resulting viscoelastic gels comprise well-developed fibrous networks, which despite having no physical cross-linking, respond as non-Newtonian fluids. These properties contrast with those of native ECMs which form highly cross-linked networks of persistent fibres that are many microns in length. This suggests that carpet-like scaffolds, the persistence for which is provided by a high density of nanoscale cylinders, should be comparably effective in supporting cell growth, which was indeed observed. Third, the carpet-like type of extracellular scaffolds provides cells with abundant focal adhesions promoting multi-polar cell morphologies with low-adhesion migration modes. Fourth, owing to the amphipathic nature of the β-helix, i.e., the clear separation of its contiguous lysine and hydrophobic ladders, the assembly is capable of encapsulating nucleic acids and mediating their intracellular delivery, and when challenged with bacterial colonisation, is able to inhibit bacterial growth. Collectively, the demonstrated properties provide an advantage of exploiting one biofunctional motif in the generation of multifunctional materials. One example is the use of the motif as a 3D scaffold that can support cell proliferation, while inhibiting bacterial colonisation, thus potentially removing the need to use antibiotics in cell culture, and promoting gene delivery, which could offer an effective testbed system to probe combined mechanotransduction and genetic effects on cell responses in 3D. Ultimately, more apparent advantages are subject to the tuneability of the motif the potential for which is shown in the provision of physical cues manifested in specific cell responses, such as multipolar cell morphologies, and which can be enhanced or inhibited by incorporating biological signals and cues whose recognition patterns by cells are well understood. This is an exciting subject of exploring the biological properties of the motif in more detail in future studies.

All in all, the multiple functions the designed β-helix supports stem from its structural parameters encoded in the sequence of its elementary units, which renders the design a self-contained biofunctional motif for exploring and exploiting mechanistic biology.

## Methods

### Peptide preparation

Peptides, SaBeH and a non-assembly control peptide (QIAALEQEIAA-LEQEIAALQ-amide), were assembled (0.1 mmole scale) on a Liberty Blue peptide synthesizer using conventional Fmoc/ᵗBu solid-phase protocols, with DIC/Oxyma Pure as coupling reagents and a Tentagel R RAM (TGR RAM) resin as a solid phase. After synthesis the peptides were cleaved from the resin (94% TFA, 3% TIS, 3% water), lyophilised and salt exchanged (TFA for HCl) to give peptide HCl salts. The resulting products were dissolved in acetonitrile (2.5 ml) and diluted into filtered (0.22 μm) MilliQ water (7.5 ml). The clear solution was then filtered through cotton wool and centrifuged (×1900, 15 min) to

remove any particulates. The peptides were purified by reversed phase high performance liquid chromatography (RP-HPLC) and their identity was confirmed by liquid chromatography mass-spectrometry (LCMS) and analytical HPLC. RP-HPLC purification was performed on a Waters, XSelect, CSH, C18 column (PN: 186005493, 30 mm × 250 mm, 130 Å, 5 μm), with detection at 280 and 214 nm. Preparative runs were performed at 40 mL/min: using a 15–55% solution B gradient over 40 min (solution A is 5% and solution B is 95% aq. $CH_3CN$, both containing 0.1% TFA). Analytical runs were performed at 1 mL/min using a 0–100% solution B gradient (over either 20 min or 50 min) using C18 (PN: 186005291, 4.6 mm × 250 mm, 130 Å, 5 μm). LCMS was performed on a Thermo Scientific Q-Exactive system (equipped with a HESI probe), using 4 μm Phenomenex Synergi C12 (00F-4337-E0) analytical columns. LCMS runs used a 0–100% solution B gradient over 20 min at 0.8 mL/min with detection at 280 and 214 nm (solution A, 5% and solution B, 95% aqueous $CH_3CN$, both containing 0.1% formic acid). Prior to lyophilisation, SaBeH was dissolved in 50 mL of aq. $CH_3CN$ (40%) containing formic acid (0.1%) and the resulting mixture was stirred over 16 h at room temperature. This additional step was performed to remove the likely Trp(carbamate) adduct, $[M + H]^+ + 44$ Da as observed by LC-MS. Following the procedure and lyophilization, the carbamate removal was confirmed by LCMS. The yields of lyophilized SaBeH by weight before and after purification were 70–80% and 5–10%, with purity by RP-HPLC being 75% and 95%, respectively.

LCMS, m/z (ESI$^+$), data is reported as m/z observed/Da (assignment = m/z expected/Da): SaBeH: 995.07 $(1/2[M + 2H]^+ = 995.07)$, 663.72 $(1/3[M + 3H]^+ = 663.72)$; a non-assembly control: 1076.56 $(1/2[M + 2H]^+ = 1076.09)$, 718.04 $(1/3[M + 3H]^+ = 717.73)$.

**Computational analysis and simulations.** Sequence and 3D structure predictions of individual β-helices were performed using AlphaFold2 as per the original protocols[5]. Once generated, three individual β-helices with W residues in each individual helix facing W residues of the other two helices were brought together to provide an initial configuration. PACKMOL was used to pack the helices together and optimise the packing and spatial region of the initial configuration[78]. Following energy minimisation and equilibration phases using canonical ensembles (fixed number of atoms, volume, temperature and pressure), the structure was simulated over 100 ns giving rise to a β-helical cylinder (Fig. 1f). MD simulations were performed over 100 ns and 200 ns to assess the stability of the cylinder. All MD simulations were performed using the GROMACS molecular dynamics package. Every MD simulation was carried out in triplicate, with the behaviour of the fold being consistent across the replicates. The CHARMM36M force-field was used to describe the proteins within our simulations[79]. Particle-Mesh Ewald and LINCS were respectively employed to set up electrostatics and reset bonds to correct lengths after an unconstrained update. The TIP3-P model was used to represent water molecules, and Na$^+$ and Cl$^-$ represented counter-ions. The cut-offs for both Lennard-Jones and Coulomb potentials were 12 Å. The potential energy was minimized by using 5000 steps of the steepest-descent algorithm and reaching the maximum force value (tolerance) smaller than 1000 kJ mol$^{-1}$ nm$^{-1}$[80]. A heating protocol, with increasing temperature 0–303.15 K at 125 ps, used the Nose-Hoover thermostat with a time integration step of 2 fs. For the isothermal-isobaric ensemble, the pressure was increased from 0 bar to 1 bar using the Parrinello-Rahman barostat and the temperature was kept constant at 303.15 K for 125 ps with a time integration step of 2 fs. The Parrinello-Rahman barostat, which is an extension of the Andersen barostat is needed to control the pressure of the simulated system. Unlike the Andersen barostat, the Parrinello-Rahman barostat supports the anisotropic scaling of the size and shape of the simulation box. For the final production stage, the simulations were run over 100 ns at 1 bar and 303.15 K with a time integration step of 2 fs. All visualizations were rendered using VMD 1.9.3 (https://www.ks.uiuc.edu/Research/vmd/)[81].

**Statistical computational analysis**
The stability of SaBeH assembly was assessed against three parameter types: RMSD, radius of gyration and distance maps. The three approaches were combined to reveal a consistent pattern confirming the stability of the assembly. Distance maps provided a two-dimensional representation where each element (i, j) corresponds to Euclidean distance measured between two α-carbon atoms of different residues (i and j) in SaBeH. Unlike binary matrices, which are characteristic of contact maps, the elements in a distance map represent a continuous range of distances, providing detailed insight into the spatial relationships between residues. RMSD was used to quantitatively measure similarities between the superimposed coordinates of Cα atoms. Radius of gyration provided a mean-square distance from the centre of the SaBeH cylinder. All three parameters were calculated using Python based libraries and data analysis packages (NumPy, pandas, Seaborn, SciPy, Matplotlib), which were used also to perform careful cleaning and postprocessing by averaging distances from the same specific regions (i.e., intra-helical spacing, intra-arms distances and inter-arm distances) of the general distance maps. The MDAnalysis library[82] was used to compute the Radius of gyration and the RMSD.

**Circular dichroism spectroscopy.** CD spectra for SaBeH solutions (100 μM, unless stated otherwise) were prepared in filtered (0.22 μm) 10 mM 3-(Morpholin-4-yl)propane-1-sulfonic acid (MOPS), pH 7.4, at room temperature, were recorded on a JASCO J-810 spectropolarimeter equipped with a Peltier temperature controller in a quartz cuvette with 0.05 cm pathlength. All measurements were taken in millidegrees using 1 nm step, 1 nm bandwidth, 1 s collection time per step and four acquisitions. Background, buffer spectra were subtracted from the raw spectra and the spectra obtained were converted to mean residue ellipticities (deg cm$^2$ dmol$^{-1}$ res$^{-1}$) by normalizing for the concentration of peptide bonds and the path length, followed by treating the spectra with the Savitzky-Golay digital filter that is used to smooth the noise without distorting the original signal. Raw and processed spectra are given in Supplementary Fig. 7. Thermal denaturation was performed at 219 nm at variable temperatures from 20 to 80 °C with 1 °C intervals, 2 °C/min ramp rate and the equilibration time of 180 s for each spectrum.

**Fourier transform infrared spectroscopy.** FTIR spectra were recorded using an INVENIO-R spectrometer (Bruker) at a spectral resolution of 4 cm$^{-1}$, scanner velocity of 20 kHz, phase resolution 32 and zero filling factor 4. Samples (60 μM) in 10 mM MOPS buffer (20 μL, pH 7.4, room temperature) were placed in a circular sampling area (r = 2 mm; l = 6 μm). A background measurement was performed for every sample spot (replicate): 64 scans for the background and 64 scans for every sample. Each measurement was done at 2 min/spot for four replicates. OPUS (version 8.7) was used to obtain the second derivatives of 1500–1700 cm$^{-1}$ spectra. The Savitzky-Golay digital filter was used to smooth the noise of the spectra and ease the separation of shoulder bands. Thermal denaturation was recorded at each time point with a 1 temp-range ramp from 25 to 80 °C with ΔT of 5 K, and the equilibration time of 120 s.

**Photon correlation spectroscopy.** Size distributions, scattering intensity and ζ-potential of the formed complexes were measured in quartz low volume cuvettes or folded capillary cells at 25 °C using Zetasizer Ultra Red (Malvern Instruments Ltd, Worcestershire, UK). Hydrodynamic radii and ζ-potential values were obtained through the fitting of autocorrelation data using Zetasizer Software (ZS Explorer), without fitting for anisotropic assemblies. Every size distribution and CAC value represents a mean of three independent preparations with each measurement consisting of a minimum of 10 recordings.

**Thioflavin T fluorescence.** SaBeH (1.56–100 μM) assembled over three hours in filtered (0.22 μm) 10 mM MOPS was mixed with ThT (final concentration of 30 μM in 10 mM MOPS) and incubated over 60 min. The fluorescence was measured from 460 to 565 nm on a SpectraMax plate reader using the excitation wavelength of 440 nm. Fluorescence intensity at 495 nm was plotted versus SaBeH concentrations to determine CAC. Every value in the plot is a mean with standard deviation of three independent experiments.

**X-ray diffraction.** The XRD data was collected on a Bruker D8 Discover diffractometer, equipped with a D8 goniometer, which was set up in the Bragg-Brentano geometry using a θ–2θ drive. Cu-Kα radiation with characteristic X-rays of λ = 1.5406 Å was used to acquire the data. The incident X-ray beam was collimated using 1° slits, with the diffracted beam collected in a scintillation detector. The specimen wafers were maintained in a fixed position by a vacuum chuck. 2θ scans of the samples were performed from 5° to 100° with a step size of 0.01° 2θ, which was used with a count time per step of 50 s. After data collection the background signals from the data were subtracted using the Bruker EVA programme. All measurements were performed at room temperature with tube settings of 40 mA 40 kV. The pre-incubated 100 μM solution (200 μL) of peptide in 10 mM MOPS (pH 7.4) was deposited on a silicon wafer and the excess was removed by blotting paper.

**Transmission electron microscopy.** Micrographs of the SaBeH assembled overnight in 10 mM MOPS (pH 7.4) were recorded using a Talos Arctica transmission electron microscope (FEI Thermo Fisher) equipped with a Ceta16M camera and operated at 200 keV, and the micrographs were analysed using ImageJ v 1.53 and OriginPro 2021b. An 7-μL aliquot of SaBeH solution was placed on glow discharged carbon-coated (mesh 200) and the buffer excess was removed by blotting paper. The resulting sample was the stained with uranyl acetate (aq. 2%, w/v) for 10 s and buffer excess was removed by blotting paper.

For the microtome analyses, cell-free and cell-laden gels were fixed using 1% (w/w) glutaraldehyde (Agar Scientific), which were diluted in phosphate buffered saline (PBS) and pre-warmed at 37 °C for 5 min. The fixative was then replaced with fresh glutaraldehyde (1 mL, 1%, v/v, in PBS) and the samples were incubated at 4 °C overnight. Low melting point agarose (Sigma Aldrich) was used to embed the samples, which were then fixed over 1 h on ice using osmium tetroxide solution (aq. 2%, v/v, Agar Scientific) in a cacodylate buffer (aq. 0.1 M) containing potassium ferricyanide (aq. 1.5%, w/v). The resulting samples were stained with uranyl acetate (aq. 2%, w/v) over 1 h at room temperature. The samples were dehydrated with ethanol at increasing concentrations (50%, 70% 80%, 90%, 100% (2×), 5 min each), which was followed by washing (10 min) at room temperature with anhydrous acetone (2×). The samples were gradually infiltrated with Durcupan resin (Sigma Aldrich), which was polymerized over 48 h at 60 °C. The prepared samples were sectioned using Leica Ultracut EM UCT (Leica Microsystems) at 70 nm and were collected on Formvar coated 200-mesh copper grids, which were then analysed using a transmission electron microscope (Talos Arctica or JEM1400-Plus, 120 keV, LaB6, equipped with a Gatan OneView 4 K camera).

**In-liquid AFM and Peak Force QNM imaging.** Peptide was diluted in filtered (0.22 μm) MilliQ water or 10 mM MOPS (pH 7.4) at room temperature. The concentration of the solutions was measured with a NanoDrop at 280 nm. The solution was diluted to 100 μM in filtered (0.22 μm) 10 mM MOPS at pH 7.4 at room temperature and assembled over two hours. The solution was then diluted further to 50 μM and placed on freshly cleaved mica before imaging.

For single cylinder imaging, the assembly solution was diluted to 1 μM before placing on freshly cleaved mica. Imaging was performed in liquid using PeakForce Tapping mode on a Multimode 8 AFM system (Bruker AXS, USA) using MSNL-E (Bruker AFM probes, USA) cantilevers (0.1 N/m spring constant and 38 kHz resonant frequency) or PeakForce-HIRS-F-B (0.1 N m-1 spring constant and 100 kHz resonant frequency) and a fluid cell (Bruker AXS, USA). Images were taken at a PeakForce frequency of 2 kHz, PeakForce amplitude of 20–50 nm and PeakForce setpoint of 100–150 pN. First-order line-by-line background subtraction was carried out using Gwyddion. Dimensional analysis of SaBeH structures was carried out by drawing line profiles through cross-section and along the structures, respectively. A Gaussian fit was applied to curves to determine diameter values, which were validated by calculating widths corrected for the finite tip size[83]. The helical periodicity was identified from the power spectra obtained by a one-dimensional Fourier transform along the peptide axis. Two pronounced peaks were observed, corresponding to the first $(\pi/\lambda)$ and second $(2\pi/\lambda)$ order wave vectors, which each yielding a helical repeat of λ = 28.5 ± 7.3 nm and λ = 15.4 ± 3.8 nm respectively.

**Hydrogel preparation.** SaBeH-laden hydrogels were prepared as 1% w/v gels, unless stated otherwise. The peptide was dissolved in water or 10 mM MOPS buffer (pH 7.4) warmed to 37 °C. Media composed of 25% v/v 10 × DMEM, 25% v/v FBS, and 1 × supplemented DMEM was warmed to 37 °C. An equal volume of cell suspension in media was added and gently mixed with the peptide. The mixture triggers peptide assembly and gelation. Collagen type I from rat tail (5 mg/ml, IBIDI, UK) was prepared according to manufacturer instructions at concentration of 1.5 mg/ml. Gels were placed in cell culture incubator (37 °C, 5% CO$_2$, 95% humidity) for 30 min to allow for gelation before topping up with 1 × supplemented cell culture media.

**Rheology.** Rheology experiments were performed for SaBeH gels (1%, 2% and 5% w/v) prepared as described in the section above−hydrogel preparation. Rheological measurements were performed using an AR-G2 rotational rheometer (TA Instruments). The specimen was held between two parallel plates to a rotational or oscillatory torque. The angular displacement response of the specimen to that torque was measured. The rheological properties were determined from the torque and displacement values and the geometry dimensions. A set of 25 mm diameter parallel plates was used with a gap between the upper and lower plates of 750 μm. The temperature was maintained at 25 °C for all the tests. Shear modulus was measured by applying a sinusoidal torque to the top plate and measuring the sinusoidal displacement. From the torque and displacement signals and the gap and plate diameter values, shear storage G′ and shear loss G″ moduli were determined. The former and the latter represent the elasticity and the viscous response of the specimen, respectively. Loss angle (δ) results were also obtained, where the tangent of the loss angle (tan δ) is equal to the ratio of the shear loss modulus G″ to shear storage modulus G′. A high loss angle is indicative of a specimen that is dominated by its viscous behaviour, whereas a low loss angle is indicative of a specimen that is dominated by its elastic behaviour. The oscillatory testing was carried out over a range of frequencies from 1 rad/s to 200 rad/s. Viscosity was measured using steady shear testing, the upper plate was rotated continuously at a given angular speed and the torque required was measured. From the torque and displacement signals and the gap and plate diameter values, the shear viscosity was determined. Steady shear testing was carried out over a range of shear rates from 1 to 100 (1/s), to assess Newtonian or non-Newtonian behaviours.

**Cell culture.** HeLa cells (ATCC; CCL-2), HeLa reporter cells expressing GFP (Cell BioLabs Inc; AKR-213), HDFs (Merck, UK; 106-05 A) and turboFP602 transfected HDF (Innoprot, Spain; P20204) were used from frozen as supplied as per the suppliers' instructions. All cell lines were tested negative for mycoplasma using a universal mycoplasma detection kit (ATCC; 30-1012k). The cells were maintained in 1× Dulbecco's

Modified Eagle's Medium (DMEM) cell culture medium supplemented with 10% v/v fetal bovine serum (FBS) and antibiotics (gentamicin and amphotericin B) at 37 °C, 5% $CO_2$, and 95% humidity. The cells above 80% confluency were washed (×3) with PBS and trypsinized followed by adding serum supplemented media to eliminate secondary toxic effects of trypsin. Detached cells were spun down by centrifugation, and the excess solvent was replaced by the cell growth media and seeded in SaBeH and collagen gels at the density of 500 cells per μL.

**Cell viability and proliferation in hydrogels.** PrestoBlue™ reagent (ThermoFisher Scientific) was added to each sample according to the manufacturer's protocol: the reagent supplied as a 10× solution was added to each well by diluting (1×) in culture medium. The cell loaded hydrogels were incubated for 120 min (37 °C, 5% $CO_2$, 95% humidity) in the diluted reagent. The fluorescence of each well was measured with a microplate reader (BMG Labtech, Germany), with 544 nm excitation and 590 nm emission filters.

LDH assays (CyQUANT LDH Fluorescence Kit, Invitrogen) were performed with medium transferred from every sample (SaBeH, NCTI, 2D matrix-free) to a 96 well plate followed by adding a reaction mixture. The plate was then incubated at room temperature for 10 min before adding a stop solution. The fluorescence was recorded at 560 nm (excitation) and 590 nm (emission).

Colorimetric 3-(4,5-dimethylthiazol-2-yl)−2,5-diphenyltetrazolium bromide (MTT) assays (CyQUANT MTT kit, Invitrogen) were performed by adding a stock solution of the reagent (final concentration of 1.2 mM) to cell-laden SaBeH gel samples, followed by incubations for 4 h (37 °C, 5% $CO_2$ and 95% humidity). The resultant formazan crystals were then dissolved in dimethyl sulfoxide according to the manufacture's protocol and were further incubated for 10 min. Absorbance intensities were recorded by a microplate reader at 540 nm. MTT was used at a range of concentrations (1.2–1200 μM) to probe the impact of concentration on the results.

Click-iT EdU (5-ethynyl-2′-deoxyridine) (ThermoFisher Scientific) at 10 μM was added to cells before seeding into gels. At defined time points (24, 48 and 72 h), gel samples were fixed with 4% paraformaldehyde for 15 min at room temperature, washed with 1% (w/v) bovine serum albumin (BSA) in PBS (×3) and incubated with 0.5% Tergitol 15-S-9 for 15 min. The EdU reaction cocktail was prepared according to the manufacturer's protocols and added to the samples to incubate over 30 min at room temperature. Samples were washed with 3% (w/v) BSA (×3) and EdU fluorescence was measured on a flow cytometer at 638 nm (excitation) and 660 nm (emission) to give EdU-positive cells.

CellTrace Violet (ThermoFisher Scientific) at 5 μM was added to cells followed up by incubating for 20 min (37 °C, 5% $CO_2$ and 95% humidity). Cells were then washed with PBS (×3) before seeding into gels. The obtained samples were measured at defined time points (24, 72, 168 h) on a flow cytometer at 405 nm (excitation) and 450 nm (emission) to give CellTrace-positive cells.

All assays were performed as per the manufacturer's protocols. Background for all assays was determined for cell-free gels and subtracted from the readout values for cell-laden gels. Data is presented for at least three independent biological replicates and is expressed in percentage of positive cells (e.g., metabolically viable) against the total number of cells, taken as 100% (EdU), a manufacturer's positive control sample, taken as 100% (LDH), the total number of metabolically viable cells grown in NCTI, taken as 100% (PrestoBlue, MTT).

**Reagents for transfection.** LF (Lipofectamine LTX Plus and RNAimax) was purchased from ThermoFisher and used as per the supplier's protocols. DNA plasmids (pkm516 and iLOV) were kindly donated by Prof J. M. Christie[84]. Plasmids were labelled using Label IT Nucleic Acid Labelling kit Cy5 from Mirus Bio. Alexa-647 labelled siRNA was custom synthesised by Eurogentec (5′→3′: GCA-AGC-UGA-CCC-UGA-AGU-UC).

SaBeH stock solutions were diluted in MilliQ water (0.22 μm) at the desired charge ratios (CR) of peptide amino groups (P) to nucleic acid phosphate groups (N). The final volume of the complexes was kept below 25 μL. Size measurements were done for 60 μL. The peptide dilutions were added to the varying amounts of plasmid DNA (100–24600 ng), or siRNA (7.5–60 pmole) mixed by pipetting, and left for 20–30 min to incubate at room temperature to complete complexation.

**Plasmid preparation.** Plasmid transformation was performed in high-efficiency NEB 5-alpha competent *Escherichia coli* cells (New England Biolabs, Ipswich, UK). Plasmid DNA (1 μL) was added to the cells and the mixture was left on ice for 30 min and then treated for 30 s at 42 °C. The mixture was placed on ice for another 5 min and super optimal broth was added (450 μL). The obtained preparation was shaken (250 rpm) over 1 hr at 37 °C. Cells were pelleted by centrifugation (1000 × *g*, 5 min), and the supernatant was aspirated and discarded. The pellet was transferred into Lysogeny Broth containing antibiotics (LB, 5 mL).

Plasmid extraction was carried out with the Plasmid Maxi kit (Qiagen, Hilden, Germany). All buffers and reagents used were supplied in the kit. The obtained transformation pellet (5 mL) was inoculated in the LB (50 mL) and incubated with shaking (250 rpm) at 37 °C overnight. The culture was then treated as per the manufacturer's protocol with supplied reagents. Following the treatment extracted DNA was air-dried over 10 min, redissolved in the deionized water (0.5 mL) and was stored at −20 °C until use.

**Transfection and gene expression assays.** Prior to transfection, GFP-expressing HeLa or HeLa cells were incubated for 24 h and washed (×3) with PBS. Prepared pDNA and siRNA complexes with SaBeH or LF as well as free pDNA and siRNA were diluted in OptiMEM® serum reduced media, then added to cells to incubate over three hours. Transfection assays were run after 3 h whereas for plasmid expression assays after the incubation, the cells were supplemented with complete DMEM media containing 10% FBS and incubated for a further 45 h. For SaBeH gel assays the peptide was assembled with an iLOV plasmid in Ibidi chambers (Thistle Scientific, UK) at 2% w/v in 10 mM MOPS buffer, pH 7.4. After 30 min, either HeLa (expression) or HeLa GFP (knockdown) cells in media were added to adjust for 1% w/v gel. Cell loaded gels were left to incubate for 30 min (37 °C, 5% $CO_2$, 95% humidity) before adding complete DMEM, which was changed every 24 h until the final time point at 48 h. Cells were then analysed by confocal fluorescence microscopy or flow cytometry. Uptake and expression efficiencies are expressed as the total fluorescent cell counts of the total cell counts (taken as 100%), unless stated otherwise.

**Cell viability during transfection.** HeLa cells incubated overnight were washed, and transfection complexes diluted in OptiMEM™ (100 μL) were added. Three-hour post transfection incubations, 100 μL of the serum supplemented DMEM were added to each well, and the plates were incubated for 24 and 48 h. PrestoBlue™ reagent (ThermoFisher Scientific) was supplied as a 10× solution and added to each well by diluting (1×) in the culture medium. The cells were incubated for 2 h (37 °C, 5% $CO_2$, 95% humidity). The fluorescence of each well was measured with a microplate reader (BMG Labtech, Germany), using 544 nm excitation and 590 nm emission filters. Standard calibration curves (200–20,000 cells) were generated by plotting measured fluorescence values versus cell numbers. Total viable cell counts are expressed in percentage, viability of control samples (untreated cells) is taken as 100%, and the data is presented for at least three independent biological replicates.

**Cell uptake inhibitors experiments.** HeLa cells were pre-treated for 30 min with Opti-MEM (100 μL) containing either 50 mM ammonium

chloride, 200 nM bafilomycin A1, or 10 mM sodium azide and 50 mM deoxyglucose. Following the incubations, the cells were washed with OptiMEM (100 µL) and 150 µL of SaBeH·DNA (CR4), HaL° (CR4), LF-DNA or free pDNA was added to the cells and incubated over 2 h before the analysis of Cy5 plasmid uptake by flow cytometry. After subtracting the uptake of the bare pDNA from the obtained data, the mean Cy5 fluorescence for each sample was expressed in percentage of median fluorescence recorded for untreated samples, taken as 100%, with data presented as averages with standard deviations for three independent biological replicates.

**Quantification of gene encapsulation efficacy.** Sybr® Gold (Invitrogen) intercalation assays were used to determine free siRNA in mixtures (CR1, 2) with SaBeH: 20-µL of siRNA (12.8 pmole) and SaBeH (188 pmole) were co-assembled for 30 min in 10 mM MOPS (pH 7.4). 200 µL of a of Sybr® Gold nucleic acid dye (1 in $10^4$ dilution, 10 mM MOPS, pH 7.4) was then added to SaBeH/siRNA to incubate for another 10 min, which was then followed by recording fluorescence on a BMG plate reader at 485 nm (excitation) and 520 nm (emission).

**Agarose gel electrophoresis.** Agarose gel electrophoresis runs for pDNA co-assembled with SaBeH or mixed with the non-assembly control at different CRs were run at 75 V for 60 min in 1.0% (w/v) agarose in Tris/borate/EDTA buffer containing 1× SYBR safe DNA Gel stain (Thermo Scientific). The gels were UV visualised and photographed by a Bio-Rad Gel Doc EZ system. DNase I (amplification grade, 1 U/µL, from Thermo) was added to peptide-pDNA samples (1 µg) and supplemented with 1× DNase I reaction buffer. The preparations were then incubated over 15 min at room temperature. Quick-Load® Purple 2-log DNA ladder was used as reference markers.

**Bacterial growth inhibition.** *E. coli* (K12) and *Staphylococcus aureus* (ATCC 29213) were treated for 60 min with SaBeH or polymixin B (Fisher Bioreagent) at 100 µM in filtered (0.22 µm) 10 mM MOPS (pH 7.4) at room temperature. Positive controls were produced by treating bacteria in 70% (v/v) aq. ethanol for 60 min. Bacteria were stained using the LIVE/DEAD™ BacLight™ bacteria viability kit (ThermoFisher, UK) according to manufacturer's protocols before placing in Ibidi chambers (Thistle Scientific, UK) for imaging or measuring by flow cytometry.

**Flow cytometry.** Quantification of uptake for labelled plasmid and siRNA and expressed fluorescent proteins were performed using a CytoFlex flow cytometer and analysed using CytExpert software V2.4 (Beckman Coulter, USA). Cells were trypsinized and supplemented with 50 µL complete DMEM media. Cy5 or protein fluorescence were measured using 638/660 nm or 488/525 nm excitation and emission filters, respectively. Uptake and knockdown results were analysed using a series of gating steps with side and forward scattering to gate cell populations first and then singlet cells from the cell gate. From the plasmid only control cells, incubated with 100 µL OptiMEM® containing no reagents, a negative cell population was selected and taken as 0% negative, allowing for false positive samples. This gate was then used on all other samples to measure the relative populations of fluorescence positive cells which were expressed in mean and median fluorescent intensities.

For bacteria tests, bacteria, treated with SaBeH or polymyxin B, used as a positive control, and untreated, used as a negative control, were numerated by flow cytometry, pre-calibrated with counting beads (Invitrogen) using proprietary protocol optimised to complement conditions used for fluorescence microscopy. Bacteria were stained with propidium iodide (PI) and SYTO 9 as a live/dead stain (Invitrogen) for 15 and 60 min, then analysed for SYTO 9 and PI using the 488/525 nm and 561/610 nm excitation and emission filters, respectively. Bacteria and counting beads were gated using forward

and side scatter, and gates for live, and polymyxin treated dead cells, were prepared for *S. aureus* (ATCC 29213) and *E. coli* (K12), respectively.

**Fluorescence microscopy.** Fluorescent images were acquired using a confocal laser scanning microscope (Stellaris 5, Leica) with 20×/0.7 NA or 63×/1.4 NA objectives and a light sheet microscope (Aurora, M2) with 10×/0.3 NA water immersion objective. Live cell imaging was performed under controlled environmental conditions (37 °C, 5% $CO_2$, 95% humidity). Approximate maxima excitation/emission at 488/509 nm and 547/602 nm were used for HeLa GFP and HDF Turbo FP602 cells, respectively. The settings were optimised for depth of the hydrogels. Images (3D stacks) were processed using ImageJ software. Cells were counted in images using a StarDist plugin 37 (https://github.com/stardist/stardist) versatile model. Segmented objects were measured in perimeter and circularity. Objects with sizes <20 µm were excluded from the analysis. Transfected cells were imaged using confocal laser scanning microscope (Stellaris 5, Leica) using a 63×/1.4 NA objective with 405/450 nm and 653/670 nm for nuclei (Hoechst 33342 dye) and siRNA /pDNA labelled with Cy5/ Alexa647, respectively. The settings were optimised for acquisition, and images (3D stacks) were processed using ImageJ software. *E. coli* (K12) and *S. aureus* (ATCC 29213) were imaged using a confocal laser scanning microscope (Stellaris 5, Leica) using a 63×/1.4 NA objective with 504/523 nm and 549/615 nm for Syto9 and propidium iodide, respectively. Images (3D stacks) were processed using ImageJ software and bacteria cells were counted for live (green) and dead (red).

Metrics such as net and total displacements, persistence, velocity and MSDs were obtained by tracking individual cells using the Track-Mate 7 plugin[85]. Image stacks were corrected for minimal imaging drifts using Linear Stack Alignment with SIFT. Cells were segmented on TrackMate using StarDist. Tracks were generated using the Linear Assignment Problem mathematical framework tracker. Non-continuous cell tracks were discarded, only continuous tracks were analysed.

**Statistical analysis.** The statistical analysis was performed using ANOVA, multi-means comparisons Bonferroni post-test and two-tailed *t* tests, unless stated otherwise, with *p* values < 0.05 or 0.01 considered significant. The results are presented as means with standard deviations.

### Reporting summary

Further information on research design is available in the Nature Portfolio Reporting Summary linked to this article.

## Data availability

The data that support the findings of this study are available within the paper and its Supplementary information files or are available from the corresponding author upon request. Source data are provided with this paper.

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

## Acknowledgements

We acknowledge funding from the UK's Department for Business, Energy and Industrial Strategy and the UK's Industrial Strategy Challenge Fund (UK's Centre for Engineering Biology, Metrology and Standards). This work was supported by the Theoretical and Computational Biophysics group, NIH Center for Macromolecular Modeling and Bioinformatics, at the Beckman Institute, University of Illinois at Urbana-Champaign. We thank the UK Materials and Molecular Modelling Hub, which is partially funded by the EPSRC (EP/P020194/1 and EP/T022213/1), and the UKHPC Materials Chemistry Consortium, which is also funded by the EPSRC (EP/R029431), for providing access to computational resources. This work also benefitted from access to the Computational Research, Engineering and Technology Environment (CREATE) at King's College London[86].

## Author contributions

C.D., J.G.R. and M.G.R. conceived and designed the study. All authors designed experiments. J.G.R. performed computational studies. C.D., J.G.R., E.H., S.R., J.E.N., A.H., A.B., I.E.K., N.F., P.A., P.G., A.T.F. and A.M. performed the experiments. J.E.N., M.S., B.W.H., C.D.L. and M.G.R. supervised the study. M.G.R. wrote the manuscript. All authors analysed data and edited the manuscript.

## Competing interests

B.W.H. holds an executive position at an AFM manufacturer Nanosurf. Nanosurf played no role in the design and execution of this study. B.W.H. has no other competing interests. The authors declare no other competing interests.
