## [Transparent Peer Review file · Nature Communications]

A self-assembled protein β -helix as a self-contained biofunctional motif

Corresponding Author: Professor Maxim Ryadnov

Version 0:

Reviewer comments:

Reviewer #1

(Remarks to the Author)

The authors report the designed and engineered construction of a protein beta-helix that self-assembles from equilateral triangular unit comprised of a core beta-strand sequence of 18 amino acid residues. A series of elegant analytical analyses confirm that the resulting discrete cylindrical nanostructure is constrained by parallel hydrogen-bonded beta-strands that results in a shared hydrophobic core. The core unit was designed to incorporate lysine residues to allow contiguous cationic ladder formation with high affinity to cellular membranes. This, in turn, enables a means for imparting functional activity. Subsequent detailed assays showed that the designed beta-helix stimulated significant cell proliferation that appeared to be mediated via its physical or morphological properties rather than by a direct biological effect. The structure also possesses bacterial inhibitory action that is likely due to its cationic properties. Finally, and unexpectedly, the beta-helix demonstrated a significant capacity to encapsulate nucleic acids and to deliver these to cells. The adoption of discrete carpet-like structures provides some basis for the observed functional effects.

Overall, the manuscript is well-written and the work exhaustively supported by a plethora of chemical and physical analyses. While the studies confirm the development of a well-defined and characterized beta-helix, it will be another matter again to more precisely engineer functional roles, for example, cellular growth inhibitory action which would be suited to cancer therapy. Yet the authors have provided the basis for rational design of such structures and further work will validate (otherwise) their strategy. Overall, this constitutes a fine example of protein engineering to yield outcomes that can potentially have significant application to de novo protein design.

The experimental detailed provided is comprehensive and appropriate and will allow others to replicate the studies.

Suggested corrections:

p. 3, line 58: Delete comma after 'turns'.

p. 5, lines 103 and 104: Delete commas after 'residue' and 'charged'.

p. 21, line 411. Delete 'for'.

Reviewer #2

(Remarks to the Author)

The manuscript entitle, "A self-assembled protein β -helix as a self contained biofunctional motif" focused on a designed β -protein. The authors describe a smart design of a β -helix, with three of them assembled in a cylindrical structure. Following the peptides preparation, the authors conducted a thorough series of experiments to provide evidence of the formation of the cylindrical structure, and to investigate its effect on the growth of human and bacterial cells. Additionally, they demonstrated that this structure mediates gene delivery. Therefore, the data presented in the manuscript has novelty and significant interest, being worth to be published in Nature Communications. As minor points to be addressed:

- Pag. 4. Figure 1, the representation includes three snapshots of the molecular dynamics (MD) studies in panels d, e and f, corresponding to the 100 ns frame. However, the methods section indicates a 200 ns simulation. Clarification is needed on why the 100 ns frame was chosen instead of the 200 ns frame.

- Pag. 9, line 172. It is indicated that the three-helix cylinder was stable during the MD simulations, referring to Fig. 4 in Extended Data. This figure only covers 100 ns, while the MD was run for 200 ns. The 200 ns figure should be included.

- Page 31. Information on the overall yield of the peptides would be valuable.
- Pag. 32. Computational analysis and simulations. The packing and molecular dynamics of the three β -helices are explained. However, it would be of interest to detail how each β -helix was generated and the protocol for its MD.
- Pag. 32. While molecular dynamics has been carried out, it would be useful to know if MD replicas have been done and whether the behavior observed was consistent across these replicas.

Reviewer #3

(Remarks to the Author)

The author has developed a new 3D scaffold using the β -helix structure, which is an interesting approach. However, it is unclear how this form of scaffold is superior to the numerous existing 3D scaffolds. Particularly, the biological experiments using this scaffold are quite sparse, and the mechanism is not addressed at all. I have attached a more detailed review below.

1. What degree of mechanical and biochemical tunability does this type of scaffold provide? Mechanically, what is the range of stiffness and viscoelasticity it offers? Biochemically, what variety of cues can it provide? Without these features, if the scaffold merely offers fixed levels of stiffness, viscoelasticity, and predefined biochemical cues, then the advantages of this scaffold are not apparent.

2. There are many methods to observe cell growth, but this study only quantifies it through a simple assay. Techniques such as EdU assay or flow cytometry for cell cycle progression should be utilized, and if the scaffold promotes cell proliferation, it is essential to elucidate the underlying mechanism. Furthermore, if the authors claim that the effect is due to physical properties since there is no binding site on the scaffold, they need to clarify which physical properties are important, for example, whether it is due to stiffness or viscoelasticity, to provide some insight.

3. In my view, neither HeLa cells nor HDF seem to be migrating significantly. Except for one or two cells shown migrating in Supplementary Figure 8d, the majority of cells do not move and remain in the same place for several hours. Therefore, if the β -helix scaffold is claimed to support cell migration, appropriate evidence needs to be provided.

4. If the β -helix scaffold also facilitates gene delivery, the reasons for this must be demonstrated. If there is speculation about cationic nanoscale encapsulators and condensation reagents in the text, at least minimal data supporting this should be presented.

Thus, what I want to emphasize are three points:

1. It is not clear what advantages this β -helix scaffold has over many other bio-scaffolds, and it is also unclear why such a complex form of scaffold needs to be created.
2. If the β -helix scaffold supports cell growth and migration, diverse methods should be used to demonstrate cell growth and migration. Currently, the data shown are too minimal. At least some data providing insight into how it supports cell growth and migration should be provided. It should be demonstrated how cell growth is promoted and how cells migrate in the absence of a cell binding motif. Additionally, if it is due to mechanical factors, which mechanical properties are involved should be clarified.
3. Insight into the mechanism that promotes gene delivery is also needed.

If the main focus of this paper is simply the development of a new scaffold and a few demonstrations, I recommend submitting to a technical journal focused on bio-scaffolds and biomaterials.

Reviewer #4

(Remarks to the Author)

This study by Ryadnov and co-workers reconstitutes a β -helix that self-assembles from an 18-amino acid sequence. Using experimental and computational methods, these assemblies form discrete nanoscale cylindrical structures. These structures act as a three-dimensional scaffold, promoting human cell growth, inhibiting bacterial growth, and facilitating intracellular gene delivery. This work would be suitable for publication after some revisions.

- LC-MS analysis of SaBeH and the non-assembly control peptide should be added into the ESI. The LC traces and the correspondent mass analyses are required.
- Please note that in the peptide purification protocol the wording buffer should be replaced by the word solution, because they are not buffers.
- Figure 2 panel c needs y axis definition and major and minor ticks on the x axis. Please also label major peaks.
- Please, embedded the legend into the Figure 2 panel d.
- Please, add the legend corresponding to the colour gradient scalebar into Figure 2 panels f and g.
- CD spectra are very smoothed for experimental conditions used. Authors should add more details regarding the acquisitions and other processing (averaging or smoothing, etc).
- CD experimental part requires more details on peptide sample preparation

- Authors should change y axis of Extended Data Figure 5 panel c to indicate clearly that it is the first derivative
- Authors should change should add a legend embedded into Extended Data Figure 5 panel a
- CD analysis on denaturation temperature should be supported by other techniques as DSC. Authors should perform at least 2 runs of heating and cooling.
- Authors should add an histogram plot with the distribution of sizes measured through TEM.
- DLS is a technique suited for spherical structures. For anisotropic ones, such the structures reported in the paper, other processing methods or specific algorithms are required. Did the authors use them? Please if so, include them in materials and methods session. Otherwise the method is not suitable and should be changed accordingly.
- Please all the concentrations values should be noted with capital M for molar (i.e. nM, μ M, etc.)
- Please check that the units correspond between CAC units in the text (12.5 nM) and in the corresponding extended figure 7B (μ M range).
- ThT assay is an appropriate procedure for the CAC calculation. Authors should perform this assay.
- Rheological data should be represented in logarithmic scale on y axis
- Stress or strain sweep analyses will be required to identify the linear viscoelastic region, along with a time sweep test.
- The sample preparation for the rheology analyses should be included in the material and methods.
- Viscosity analysis would benefit to be performed at lower shear rate in order to see at which values the gel is linearly stable.
- MTT assay at different concentrations spanning log scale is needed.
- The space between unit and number is needed throughout the manuscripts (e.g., 25 °C).

Reviewer #5

(Remarks to the Author)

Version 1:

Reviewer comments:

Reviewer #2

(Remarks to the Author)

The authors have revised the manuscript entitle, "A self-assembled protein β -helix as a self contained biofunctional motif". While several points have been addressed, there are still some aspects that required further clarified:

- In the respond the authors indicated that they performed three 100 ns replicas, and extended one of the simulations to 200 ns. However, this is not comment on the main manuscript, and can led to misunderstanding, as, for example, Fig. 1 represented a 100 ns MD frame, whereas page 6, line 128, references the 200 ns. It would be more suitable to include the 200 ns frame in Fig 1 or provide a justification for using the 100 ns frame instead. Additionally, in the RMSD plot provided in the supporting information (Fig. 5), a slight increase in RMSD is observed after 125 ns. It would be helpful to address whether this increase is a global trend, or if specific parts of the system are contributing more significantly to it.
- Regarding the yield of the peptides, while the yield prior to purification has been included, the purity of the peptides should also be reported, both before and after purification (the yield will also be of interest).
- The clarification regarding the three replicas and their similar behavior is appreciated. However, it would be valuable to include a plot over time of key distances or other relevant parameters, to highlight the reproducibility of the simulations.

Reviewer #3

(Remarks to the Author)

The authors have made an effort to address the comments provided by this reviewer. While some points have been partially addressed, certain aspects remain unclear or require further investigation. Below are the major concerns:

Cell migration

The authors have demonstrated cell migration in Supplementary Movies 1 and 2; however, the quantification provided remains inadequate. The travel distance per cell as shown in Supplementary Figure 11e is insufficient for a thorough analysis. It is essential that the authors include additional metrics such as the average migration speed and mean squared displacement. The videos depict minimal cell movement, which raises questions about the reported average travel distance of approximately 30 μ m per cell. This reviewer is unclear how such a travel distance corresponds to the minimal movement observed in the videos. Additionally, although the authors claim migration through lamellipodia or filopodia, the current videos and images do not robustly support this. Figure 4g, which displays filopodia morphology, is only a still image and does not convincingly demonstrate active migration. Cells displaying similar morphologies could be stationary and might employ different modes of migration. Therefore, the authors should provide close-up imaging of actively migrating cells to effectively illustrate this behavior.

Cell-matrix sensing

Figure 4a demonstrates that cells can indeed interact with SaBeH. However, the mechanism or binding site through which cells adhere to this scaffold remains unclear. The authors are encouraged to discuss how cells directly bind to the scaffold. Additionally, the statement, "Since SaBeH shares physical properties with other viscoelastic cell-supporting hydrogels based on native ECM proteins such as NCTI, synthetic peptides, polysaccharides, and polymers,⁴⁶⁻⁵³ the observed cell responses cannot be merely due to the bulk properties of the scaffold such as stiffness or viscoelasticity," is overly simplistic. It is important to acknowledge that, beyond physical properties, the biochemical properties of SaBeH can significantly influence cellular responses. Therefore, it is unclear whether the observed responses are driven by the physical/morphological properties (e.g., the carpet-like micro-/nano-structure of SaBeH) or its biochemical characteristics. A discussion on this distinction would be valuable.

Scaffold morphology

The authors claim that the structural properties of the scaffold influence cell behavior. If this is the case, how does the scaffold structure change with gel concentration? In Supplementary Figure 9, stiffness increases nearly threefold when the concentration increases from 2% to 5%. Does this change result in a denser packing of the scaffold, or does it alter the structure in other ways? Using SEM imaging to examine how scaffold morphology changes with concentration would help clarify whether and how structural and mechanical properties are interrelated.

Reviewer #4

(Remarks to the Author)

I would like to thank the authors for their responses to the first round of comments. The revised manuscript has improved in several parts and the authors have addressed many of the concerns raised in the first review. The manuscript is now suitable for publication. I have just one minor comment:

- I would like to suggest that the authors add the units in the Abs spectrum of Figure 2c.

Reviewer #5

(Remarks to the Author)

Version 2:

Reviewer comments:

Reviewer #2

(Remarks to the Author)

The authors of the manuscript entitle "A self-assembled protein α -helix as a self contained biofunctional motif" have adequately addressed all the issues raised. Therefore, I recommend it for publication.

Reviewer #3

(Remarks to the Author)

The authors have addressed the points raised by this reviewer.

Point-by-point responses to the Editorial and Reviewers' comments

Editorial

The manuscript has undergone a major revision as per the Editorial and Reviewers' comments. Specific changes in the main body of the manuscript are highlighted by a yellow background in an annotated pdf file "Related manuscript file". Please also note that *Extended Data Figures* were renamed to *Supplementary Figures* to comply with the Journal's format.

Reviewer 1:

This Reviewer has given a comprehensive overview of the study, acknowledged that the work was exhaustively supported by a plethora of chemical and physical analyses, commented on the impact of the work, and offered minor corrections which we have addressed as follows:

Point 1 – "p. 3, line 58: Delete comma after 'turns'.
p. 5, lines 103 and 104: Delete commas after 'residue' and 'charged'.
p. 21, line 411. Delete 'for'."

Response – as advised, all appropriate corrections were made, and the manuscript was screened for similar corrections.

Reviewer 2:

This Reviewer has supported the publication, commenting on a thorough series of experiments conducted for the study, its novelty and significant interest for the Journal, and have raised several minor points, which we have addressed as follows:

Point 1 – "Pag. 4. Figure 1, the representation includes three snapshots of the molecular dynamics (MD) studies in panels d, e and f, corresponding to the 100 ns frame. However, the methods section indicates a 200 ns simulation. Clarification is needed on why the 100 ns frame was chosen instead of the 200 ns frame."

Response – as shown in Figure 1 all simulations were done for 100 ns. The methods section was corrected accordingly. We apologise for the confusion. An additional RMSD plot for 200 ns was added in new Supplementary Figure 5b to demonstrate that the simulations over 200 ns were comparable with those over 100 ns.

Point 2 – "Pag. 9, line 172. It is indicated that the three-helix cylinder was stable during the MD simulations, referring to Fig. 4 in Extended Data. This figure only covers 100 ns, while the MD was run for 200 ns. The 200 ns figure should be included."

Response – indeed, Extended Data Figure 4 (new Supplementary Figure 5 in the revision) covers 100 ns. Methods section and the caption of the Figure were corrected. We apologise for the confusion. An additional RMSD plot showing that the cylinder remained stable over 200 ns was added in the Figure.

Point 3 – "Page 31. Information on the overall yield of the peptides would be valuable."

Response – as advised, the overall yield of the peptides was included in the revision in the methods section. In addition, HPLC traces and mass spectra for the peptides are given in the new Supplementary Figure 1.

Point 4 – "Pag. 32. Computational analysis and simulations. The packing and molecular dynamics of the three β -helices are explained. However, it would be of interest to detail how each β -helix was generated and the protocol for its MD."

Response – Alphafold2 was used to generate each β -helix as per the original, published protocols by DeepMind. As advised, the protocol given in the methods sections for generating and simulating the structure was modified in the revision to make this clearer.

Point 5 – “Pag. 32. While molecular dynamics has been carried out, it would be useful to know if MD replicas have been done and whether the behavior observed was consistent across these replicas.”

Response – every simulation was performed at least in triplicate with all replicas given consistent results. The corresponding methods section was modified to reflect this.

Reviewer 3:

This Reviewer has acknowledged the study as an interesting approach and has raised several points, which we have addressed as follows:

General point – “The author has developed a new 3D scaffold using the β -helix structure, which is an interesting approach. However, it is unclear how this form of scaffold is superior to the numerous existing 3D scaffolds. Particularly, the biological experiments using this scaffold are quite sparse, and the mechanism is not addressed at all. I have attached a more detailed review below.”

Response – although the study probes the designed motif for 3D culture, it does not focus on introducing a new or superior 3D scaffold. The study introduces a self-assembling β -helix as a biofunctional motif whose physical properties such as fibre formation, gelation, and encapsulation – all of which derive from the β -helix self-assembly – were probed for biological applications, of which the use as a cell-supporting scaffold is an exemplar.

We agree with the Reviewer on that additional biological experiments will benefit the study, and therefore more biological assays were performed as responded to the specific points raised by the Reviewer.

Point 1 – “What degree of mechanical and biochemical tunability does this type of scaffold provide? Mechanically, what is the range of stiffness and viscoelasticity it offers? Biochemically, what variety of cues can it provide? Without these features, if the scaffold merely offers fixed levels of stiffness, viscoelasticity, and predefined biochemical cues, then the advantages of this scaffold are not apparent.”

Re-emphasised **point** – “It is not clear what advantages this β -helix scaffold has over many other bio-scaffolds, and it is also unclear why such a complex form of scaffold needs to be created.”

Response – the study introduces a designed self-assembled β -helix with the comprehensive account of its properties across multiple length scales. When used in 3D culture as a scaffold, the β -helix exhibits physical properties similar to those of scaffolds derived from peptide, protein, polysaccharide, and synthetic polymers, including those with reversible covalent bonds (e.g., cited Ref: 46-53). This β -helix-based scaffold is self-assembled, which makes it analogous to the ECMs – also self-assembled and protein-based scaffolds. However, unlike the ECMs and other scaffolds which form complex and extended supramolecular or fibrous networks, this β -helix assembles into a carpet-like scaffold of short cylindrical structures. This form has yet to be reported, and its properties including biological are not known. Therefore, biological insights on how cells respond to this unknown carpet-like form as is, without additional biology or tuned properties, in terms of proliferation, morphological changes and migration were addressed by the experiments performed in this study.

Nonetheless, to expand on these findings additional data is provided in the revision including complementary biophysical characterisation, from folding to bulk properties such as impact of stress on viscoelastic properties, biological assays at the metabolic, cytotoxicity and cell division levels and real-time cell tracking for cell migration – please refer to new relevant data in Figures 3d,e, and Supplementary Figures 9, 10, 11e,f, 12c-e. and Movies 1 and 2. Please also refer to corresponding changes in the text, also given in responses to points 2-4 below.

A more detailed comparative study focusing on specific advantages of the scaffold over other existing scaffolds could indeed be of interest. There is also scope for tuneability. However, such a type of study falls under a remit of a more specialist journal and merits a separate endeavour. To reflect this, the Discussion section was modified to read:

Collectively, the demonstrated properties provide an advantage of exploiting one biofunctional motif in the generation of multifunctional materials. One example is the use of the motif as a 3D scaffold that can support cell proliferation, while inhibiting bacterial colonisation, thus potentially removing the need to use antibiotics in cell culture, and promoting gene delivery, which could offer an effective testbed system to probe combined mechanotransduction and genetic effects on cell responses in 3D. Ultimately, more apparent advantages are subject to the tuneability of the motif the potential for which is shown in the provision of physical cues manifested in specific cell responses, such as multipolar cell morphologies, and which can be enhanced or inhibited by incorporating biological signals and cues whose recognition patterns by cells are well understood. This is an exciting subject of exploring the biological properties of the motif in more detail in future studies.

Point 2 – *“There are many methods to observe cell growth, but this study only quantifies it through a simple assay. Techniques such as EdU assay or flow cytometry for cell cycle progression should be utilized, and if the scaffold promotes cell proliferation, it is essential to elucidate the underlying mechanism. Furthermore, if the authors claim that the effect is due to physical properties since there is no binding site on the scaffold, they need to clarify which physical properties are important, for example, whether it is due to stiffness or viscoelasticity, to provide some insight.”*

Re-emphasised **point** – *“If the β -helix scaffold supports cell growth and migration, diverse methods should be used to demonstrate cell growth and migration. Currently, the data shown are too minimal. At least some data providing insight into how it supports cell growth and migration should be provided. It should be demonstrated how cell growth is promoted and how cells migrate in the absence of a cell binding motif. Additionally, if it is due to mechanical factors, which mechanical properties are involved should be clarified.”*

Response – as advised, additional tests and assays were performed to probe cell viability including metabolic (MTT), cytotoxicity (LDH release), proliferation by cell division (EdU,) metabolic long-term dye retention (CellTrace Violet) and distance tracking imaging assays, including measurement done by flow cytometry. Please refer to the new Supplementary Figure 10 and corresponding new methods sections.

The corresponding text was modified to read:

The tendency was confirmed by MTTTM cell viability assays, which gave higher numbers of metabolically active cells for HDFs when compared to HeLa cells grown in the SaBeH gels (Supplementary Fig 10a). Cell viability measured by MTTTM assays was comparable to that measured by PresoBlueTM, while the results did not depend on the concentration of the tetrazolium (MTT) dye, suggesting that SaBeH had no effect on its conversion (Supplementary Fig 10b). Further evidence for SaBeH supporting functional 3D cell culture was sought in Lactate dehydrogenase (LDH) release assays. Unlike metabolic assays, these assays measure the levels of an endogenous cytosolic enzyme, i.e., LDH, which is released upon damage to the cell membrane and therefore provide a straightforward probe for cytotoxicity. The assays showed that SaBeH outperformed NCTI in providing non-toxic environment for HDFs, which was in good agreement with the tendency for SaBeH to promote the growth of the cells over HeLa cells established by the metabolic assays (Supplementary Figure 10c). To this end, all the three assays used rely on enzymatic conversions of reporter molecules or metabolites providing quantitative insights into the metabolic responses of the cells to their extracellular environments, i.e., scaffolds. Strictly speaking, such responses are secondary to proliferation rates defined at the genetic level in particular during DNA synthesis. With this in mind, proliferation assays using (5-ethynyl-2'-deoxyridine) (EdU) were performed. EdU is incorporated into newly synthesized DNA in dividing cells thereby providing a direct measure of cell proliferation.

Monitored by flow cytometry, the assays confirmed consistently high proliferation rates for both cell types over 72 hours of incubation in SaBeH. The results strongly support that SaBeH did not affect the DNA replication phase, which can serve as a checkpoint control of proliferation, at any point of cell division (Supplementary Figure 10d,e). No appreciable difference in cell proliferation could be ascertained between HDF and HeLa by the assay either. This is in contrast to that observed by metabolic assays, indicating that SaBeH had no impact on cell division. Preferences observed by metabolic assays towards supporting the growth of HDFs more than HeLa cells are likely to result from the differences in cell membranes of the two cell types. Characteristic of cancer cells, HeLa membranes are more anionic than HDF membranes and are more prone to binding to cationic SaBeH, which may impact on their proliferation.

A metabolic assay based on long-term dye retention, such as CellTrace Violet, can provide complementary evidence (Supplementary Fig 10f,g). Monitored by flow cytometry this assay is designed to relate cell divisions, with subsequent divisions giving larger cell numbers, with fluorescence intensity, which decreases with every division. Higher fluorescence intensities were recorded for HDFs at any given time point measured over 168 hours (Supplementary Fig 10f). A general tendency of decreasing intensity was observed for both cell types. However, intensities and cell numbers for HeLa were comparable for 24 and 72 hours. This is intriguing as sharp increases in cell numbers were observed for HeLa at each subsequent time point, while median fluorescence for 168 hours was found to be nearly 3-fold lower than that for 72 hours (Supplementary Fig 10f,g). With greater cell counts recorded for HeLa in 2D without matrices (Supplementary Fig 10g), this may suggest a comparatively lagged proliferation of HeLa cells in SaBeH resulting from the need for cells to adjust in 3D, e.g., through focal adhesions, spreading and clustering, the patterns of which are defined by the scaffolds, and which was indeed observed (Fig 4). Similarly, drastic increases in cell numbers were observed for HDFs between 24 and 72 hours which may also suggest cell adjustment in SaBeH (Supplementary Fig 10g). Further, cell numbers at 168 hours for HDFs and HeLa cells were comparable in SaBeH, whilst differences in median fluorescence between 72 and 168 hours for HDFs were found to fit an ideal scenario involving several cell divisions each contributing to decreasing fluorescence (Supplementary Fig 10f,g).

Taken together the results of cell proliferation and viability assays indicate that SaBeH supports cell development effectively at all genetic and metabolic levels without cell toxicity or inhibition but is likely to introduce spatial constraints that require cells to adjust to, which may manifest in differences in cell morphologies. Indeed, the morphologies of individual cells proved to be notably distinctive in SaBeH when compared to those observed in NCTI (Fig 4a,b).

The cell-supporting effect of the scaffold is indeed due to its physical properties as there is no known biology incorporated into the scaffold. It is important to re-emphasise that since the designed β -helix assembles into a carpet-like form of short nanoscale cylinders, which is different from other scaffolds, the impact of the scaffold on cell response needs to be first studied using the scaffold as is.

And indeed, clear differences in cell morphology observed between this scaffold and native collagen already demonstrate that the effect must derive at the microstructural level of the scaffold. To re-emphasise these points the corresponding text was modified to read:

Therefore, the basis of cell responses to the SaBeH scaffold must be due to its physical or morphological properties. Since SaBeH shares physical properties with other viscoelastic cell-supporting hydrogels based on native ECM proteins such as NCTI, synthetic peptides, polysaccharides, and polymers,⁴⁶⁻⁵³ the observed cell responses cannot be merely due to the bulk properties of the scaffold such as stiffness or viscoelasticity. Differences in cell behaviour are manifested in clear changes in the morphologies of the same cells seeded in NCTI and SaBeH, and therefore must be based on cell interactions with the microstructure of the scaffold, i.e., dense, carpet like matrix of SaBeH versus long-range networks of NCTI (Fig 4e, f).

Point 3 – *“In my view, neither HeLa cells nor HDF seem to be migrating significantly. Except for one or two cells shown migrating in Supplementary Figure 8d, the majority of cells do not move and remain*

in the same place for several hours. Therefore, if the β -helix scaffold is claimed to support cell migration, appropriate evidence needs to be provided.”

Response – indeed, cell migration over the first several hours was not deemed significant as stated in the original text:

“Although the location of HeLa cells in 3D did not change significantly over several hours, the migration of individual cells in SaBeH was evident (Fig 4h and Extended Data Fig 11d), which together with blebbing observed for the cells suggests an amoeba-like behaviour, typical of cancer cells (Extended Data Fig 11d).⁵⁴”

However, it should be noted that this is under the assumption that it occurs along 2D trajectories of travel without adding up migrations in 3D at different depths, and that cells are perceived to move long distances, which is not a given, not normally observed and does not fully define cell migration.

Critical manifestations of cell migration include the leading-edge extensions – indeed, profusely expressed lamellipodia and filopodia were observed in SaBeH; cell adhesion to the matrix; indeed, distinctive multipolar and spread morphologies were observed for both cell types; cell-cell interactions – indeed, cell clustering leading to spheroid formation; release from contact sites, i.e., cell motility – indeed, observed for individual cells covering 10s of microns distances in 2D alone.

In addition, migration depends on a number of factors including a proper interpretation of external cues by migrating cells. Since there is no known biology incorporated into the scaffold the cues the cells recognise are physical, but not necessarily normal for cells, which reflects in morphological changes of the cells and their lagged proliferation resulting from the need to adjust in the SaBeH environment. This effectively suggest a potential for tunability for the design at the expense of incorporating biological signals and cues recognisable by the cells.

Nonetheless, additional cell tracking tests are included in the revision – please refer to Supplementary Movies 1 and 2, and Supplementary Fig 11e,f, showing 2D travel distances at up to 80 microns for individual cells of <20 microns in size comparable to those reported by others (new Ref: 58, 59).

To make these points clearer the corresponding text was modified to read:

Although changes in the location of individual cells in 3D at different depths are not deemed straightforward to determine unambiguously and did not seem to be significant for HeLa cells in SaBeH over several hours, the migration of individual cells was evident (Fig 4h and Supplementary Fig 11d-f). The real-time tracking of migrating cells in the same field of view gave travel trajectories covering distances of up to 80 μm over the first 5-10 hours (Supplementary Fig 11 e,f and Supplementary Movies 1 and 2). Such distances and migration rates, even without adding up travel trajectories in 3D at different depths, were comparable to those of individual and collective cell migration in 2D and 3D reported by others.^{58,59} It should be noted, however, that critical manifestations of cell migration are complex and include the leading-edge extensions, i.e., abundant lamellipodia and filopodia observed in SaBeH; cell adhesion to the matrix, i.e., distinctive multipolar and spread morphologies observed for cells in SaBeH; cell-cell interactions, i.e., cell clustering leading to spheroid formation; and release from contact sites, i.e., cell migration at distances over 10s of microns in 2D alone.

and

Since cell migration depends on a number of factors including a proper interpretation of external cues by migrating cells and since there is no known biology incorporated into SaBeH the cues the cells recognise are physical, but not necessarily normal for cells, which reflects in morphological changes of the cells and their lagged proliferation resulting from the need to adjust in the SaBeH environment. This effectively suggest a potential for tunability for the design at the expense of incorporating biological signals and cues recognisable by the cells.

Point 4 – “If the β -helix scaffold also facilitates gene delivery, the reasons for this must be demonstrated. If there is speculation about cationic nanoscale encapsulators and condensation reagents in the text, at least minimal data supporting this should be presented.”

Re-emphasised **point** – *Insight into the mechanism that promotes gene delivery is also needed. If the main focus of this paper is simply the development of a new scaffold and a few demonstrations, I recommend submitting to a technical journal focused on bio-scaffolds and biomaterials.*”

Response – as advised, additional data to provide more mechanistic insights into gene delivery is provided in the revision. This includes experiments with cell uptake inhibitors such as ammonium chloride, antibiotic bafilomycin, sodium azide and deoxyglucose – which are used to elucidate the likely mechanism of cell uptake; SybrTM Gold intercalation assays and DNA retardation gels to quantify encapsulation efficacy. Please refer to the new Supplementary Figure 12c-e and corresponding methods in the methods section.

The exact mechanism whereby SaBeH promotes intracellular delivery remains to be elucidated and merits a separate study. Nonetheless, one can speculate that with SaBeH lacking known specialist biology, such a mechanism is likely to be similar to those employed by cationic nanoscale encapsulators and condensation reagents. Like the reagents of this type, SaBeH is a nanoscale, amphipathic and cationic agent, which effectively packages and delivers genetic material into the cell without cytotoxicity (Fig 5 and Supplementary Figure 12). To gain a better mechanistic insight into the intracellular delivery of genetic material by SaBeH, cells before transfections were treated with cell uptake inhibitors, and the results obtained were compared with transfections without the treatments and transfections performed using a cationic encapsulator (LF) and a cationic condensation reagent (HaL^o). As gauged by flow cytometry, preincubations with NH₄Cl, which blocks endosomal acidification and consequently egress, caused 20-30% decreases in fluorescence when compared to that of untreated cells. With similar decreases observed for LF and no decreases observed for HaL^o, with both reagents known to undergo endocytosis, the results indicate that SaBeH may remain entrapped in the endosomes, or its endosomal escape is independent of acidification. However, pre-treatments with antibiotic bafilomycin A1 (Baf A1), which arrests both endosomal uptake and acidification, had no effect on SaBeH-promoted transfections (Supplementary Fig 12c). By contrast, reduced uptake was observed by the other two reagents which is indicative of inhibited both endosomal uptake and escape. The same levels of cell uptake by SaBeH with and without the treatment of Baf A1 may suggest a non-endosomal route of cell entry. However, the effect by Baf A1 relies on the inhibition of ATPase proton pumps regulating sub-membranous pH, which relates to energy dependence, as well as the formation of macropinosomes. To test if the cell entry was energy-dependent, the cells were pre-treated with inhibitors of ATPases – sodium azide and deoxyglucose. Similar levels of reduction in the uptake were observed for SaBeH-transfected cells pretreated with NH₄Cl, which is consistent with an energy-dependent uptake characteristic of peptide transfection reagents, such as HaL^o, and in contrast to energy-independent pathways common for liposomal reagents,⁶⁷ such as LF (Supplementary Figure 12c). Collectively, the results indicate that SaBeH exhibits an energy-dependent and acidification-independent cell uptake with no dependence or limited capacity for endosomal egress. The latter can also be impacted by the efficacy with which SaBeH encapsulates nucleic acids. SybrTM Gold intercalation assays, which are used to detect free nucleic acids and hence quantify their encapsulated forms, gave nearly quantitative gene encapsulation (up to 94%) at low CRs (Supplementary Figure 12 d). DNA retardation gels provided complementary evidence for this. No electrophoretic mobility was observed for SaBeH co-assembled with pDNA at the CRs whereat the encapsulation was essentially complete. This was in contrast to pDNA used bare and mixed with the non-assembly control peptide even at higher CR4, which exhibited the electrophoretic mobility expected for its size (Supplementary Figure 12e). In addition, these preparations did not survive treatments with DNase I whereas the enzyme had no apparent effect on pDNA co-assembled with SaBeH confirming that pDNA was protected by encapsulation (Supplementary Figure 12e). Consistent with their positive ζ -potential values, the SaBeH assemblies

remained cationic under the conditions used, and thus delivering genetic cargo into the cell as nanoscale, cationic agents.

Reviewer 4:

This Reviewer has supported the publication, given a succinct overview of the work, and advised some revisions, which we have addressed as follows:

Point 1 – “LC-MS analysis of SaBeH and the non-assembly control peptide should be added into the ESI. The LC traces and the correspondent mass analyses are required.”

Response – as advised, the mass-spectra and HPLC traces were added in the revision. Please refer to the new Supplementary Figure 1. Please also note this in addition to the LC-MS analysis in the methods section.

Point 2 – “Please note that in the peptide purification protocol the wording buffer should be replaced by the word solution, because they are not buffers.”

Response – as advised, the correction was made in the revision in the methods section.

Point 3 – “Figure 2 panel c needs y axis definition and major and minor ticks on the x axis. Please also label major peaks.”

Response – as advised, Fig 2c was revised to include the Y axis definition, major and minor ticks on the X axis and labels for major peaks.

Point 4 – “Please, embedded the legend into the Figure 2 panel d.”

Response – as advised, the legends were embedded into Fig 2d.

Point 5 – “Please, add the legend corresponding to the colour gradient scalebar into Figure 2 panels f and g.”

Response – as advised, the legends were added to the colour (height) scale bars in Fig 2 f, g.

Point 6 – “CD spectra are very smoothed for experimental conditions used. Authors should add more details regarding the acquisitions and other processing (averaging or smoothing, etc).”

Response – as advised, the corresponding section was modified to include details of the data processing and smoothing used for the CD spectra. In addition, Supplementary Figure 5 (new Supplementary Figure 6 in the revision) was extended to include raw spectra before smoothing.

Point 7 – “CD experimental part requires more details on peptide sample preparation.”

Response – as advised, details on peptide sample preparation was added in the methods section. Please also note that the details – assembly conditions – were given in the caption of Fig 2.

Point 8 – “Authors should change y axis of Extended Data Figure 5 panel c to indicate clearly that it is the first derivative.”

Response – as advised, the units in the Y axis of Extended Data Figure 5c (new Supplementary Fig 6c in the revision) were corrected.

Point 9 – “Authors should change should add a legend embedded into Extended Data Figure 5 panel a.”

Response – as advised, the legends for intermediate spectra were embedded in Extended Data Figure 5a (new Supplementary Fig 6a in the revision).

Point 10 – “CD analysis on denaturation temperature should be supported by other techniques as DSC. Authors should perform at least 2 runs of heating and cooling.”

Response – thermal denaturation monitored by CD spectroscopy is performed to probe conformational preferences and a two-state transition between the folded and unfolded forms. The stability and

reversibility of the fold is inferred by recording the spectra after the melt. A second run of heating and cooling is not deemed necessary as it does not probe the stability of the fold from its native folding conditions and is reflective of its secondary recovery from its denatured form.

Nonetheless, we repeated thermal melts with two runs of heating and cooling – (new Supplementary Figure 6e) – to reflect this.

Similarly, another method to support the data should be comparable to CD spectroscopy in the ability to probe the conformation preferences and two-state transition of the fold. With this in mind, thermal denaturation under the same conditions was performed by FT-IR spectroscopy which revealed a comparable transition midpoint – Supplementary Figure 6f.

The corresponding text was modified to read:

Moreover, CD spectra for repeated runs of heating and cooling recorded for different sample preparations at the same concentrations confirmed not only the reversibility of the fold, but also its quantitative recovery following thermal denaturation (Supplementary Fig 6e). Thermal denaturation experiments performed under the same conditions using FT-IR spectroscopy to provide complementary evidence for the conformational properties of the fold returned a comparable T_M of ~67 °C (Supplementary Fig 6f).

Point 11 – “Authors should add an histogram plot with the distribution of sizes measured through TEM.”

Response – as advised, a histogram plot showing size distributions was added to Extended Data Figure 6 (new Supplementary Figure 7 in the revision).

Point 12 – “DLS is a technique suited for spherical structures. For anisotropic ones, such the structures reported in the paper, other processing methods or specific algorithms are required. Did the authors use them? Please if so, include them in materials and methods session. Otherwise the method is not suitable and should be changed accordingly.”

Response – DLS relates Brownian motion to the size of a particle in solution. The diameter of the particle reflects the way the particle diffuses in its liquid environment, i.e., it gives a hydrodynamic diameter – which is what is reported in the manuscript. The hydrodynamic diameter is an equivalent spherical size and does not require for the object to be spherical.

For a non-spherical particle, it is assumed as the diameter of a sphere with the same translational diffusion speed as the particle. For anisotropic particles, changes in length will affect the size, but changes in diameter are harder to detect as these would have no effect on the diffusion speed.

Since DLS does not capture the shape of a particle, irrespective of the processing method used, DLS data are discussed together with the results obtained by TEM (fixed samples) and AFM (samples in solution) measurements.

To make these points clearer the corresponding text was modified to read:

DLS measurements relate the Brownian motion of a particle to its diameter by assigning to it a hydrodynamic diameter, which for a non-spherical particle assumes the diameter of a sphere with the same translational diffusion speed as the particle. For anisotropic particles, such as the observed cylinders, changes in length will affect the size determined by DLS, but changes in diameter are harder to detect as these would have no effect on the diffusion speed. Since DLS does not capture the shape of a particle the data obtained is to complement the results of the TEM and AFM measurements which provide more accurate and direct determination of size and shape.

No additional processing was used. A note to this effect was added in the methods section.

Point 13 – “Please all the concentrations values should be noted with capital M for molar (i.e. nM, μ M, etc.)”

Response – as advised, all concentrations were checked for capital M and corrected where appropriate. Please also note that in Fig 3c 50 μ m refers to thickness rather than concentration. The caption was modified to make it clearer.

Point 14 – “Please check that the units correspond between CAC units in the text (12.5 nM) and in the corresponding extended figure 7B (μ M range).”

Response – as advised, this was corrected in the text. We apologise for the confusion.

Point 15 – “ThT assay is an appropriate procedure for the CAC calculation. Authors should perform this assay.”

Response – as advised, ThT fluorescence assay was performed to confirm CAC determined by DLS. The data is presented in the new Supplementary Figure 8d in the revision, a new methods section was added in the revision and the corresponding text was modified to read:

Thioflavin T (ThT) fluorescence assays performed to complement the DLS data effectively gave the same CAC (Supplementary Fig 8d). As the fluorescence of ThT increases upon its binding to β -sheet assemblies, titrations of SaBeH into a ThT solution allowed to plot fluorescence changes versus SaBeH concentrations to confirm the CAC determined by DLS (Supplementary Fig 8c, d).

Point 16 – “Rheological data should be represented in logarithmic scale on y axis.”

Response – as advised, the data is represented using log scale on Y axis. Please also note additional data is presented in the revision (new Supplementary Figure 9) for 1%, 2% and 5% gels, and Fig 3 d,e was modified to include an oscillatory sweep analysis.

Point 17 – “Stress or strain sweep analyses will be required to identify the linear viscoelastic region, along with a time sweep test.”

Response – as advised, to identify the linear viscoelastic region oscillatory sweep analysis was included in the revision, together with the corresponding methods section. Please note the analysis plot is in Fig 3e.

Point 18 – “The sample preparation for the rheology analyses should be included in the material and methods.”

Response – as advised, the sample preparation for rheology was included in the methods sections.

Point 19 – “Viscosity analysis would benefit to be performed at lower shear rate in order to see at which values the gel is linearly stable.”

Response – the gels exhibited a classical shear-thinning at 1-100 s^{-1} , characteristic of peptide gels and polymer gels. Since the properties of SaBeH gels are defined by its fibrous microstructure which lacks physical crosslinks, the shear-thinning behaviour can be expected at lower shear rates. Indeed, this was observed. At lower shear rates the behaviour did not change as expected for a non-Newtonian fluid. Please refer to the new Supplementary Fig 9b (1% gel).

The corresponding text was modified to read:

Complementary to this, the storage (G') and loss (G'') moduli of 1, 2 and 5% (w/v) gels measured in the 0.1–100 rad/s frequency range were characteristic of soft gels including protein and β -structured gels (Fig 3d and Supplementary Fig 9a).⁴⁶⁻⁴⁸ No cross-over between G' and G'' was observed for the gels suggesting that the gel remained largely elastic over the frequency range and concentrations used.⁴⁹ The elasticity of the gels can be attributed to its dense fibrous structure dominating its mechanical properties.⁴⁶ At high frequencies (>30 rad/s), $\tan \delta$ loss decreased indicating increased elasticity due to the dynamic stiffening of the gel microstructure (Fig 3d and Supplementary Fig 9a).⁴⁷ However, contrary to linearly elastic hydrogels, the SaBeH gel exhibited a more complex behaviour.

When a shear force was applied, the gel responded as a non-Newtonian fluid demonstrating a strong shear-thinning behaviour manifesting in the gradual loss of viscosity for all gel concentrations as the shear rate increased (Supplementary Fig 9b). This effect is consistent with the anisotropic assemblies of SaBeH aligning along the direction of the shear force, resulting in the increased free space between them and reduced viscosity of the gel. This is also intriguing as for SaBeH G' was greater than G'' (Fig 3d and Supplementary Fig 9a), suggesting a well-developed fibrous network in the gels, whereas the two moduli remained relatively low (<1 kPa) in the frequency range used, indicating that the gels had a low density of physical cross-linking in its microstructure. The same behaviour held true for 5 % gels, for which marginally increased G' (≥ 1 kPa) and $\tan \delta$ loss decreasing at higher frequencies (>60 rad/s) suggest increased fibrous density (Supplementary Fig. 9a). Furthermore, oscillation stress sweeps performed to probe the microstructure of the gel revealed that the linear viscoelastic region of the gel was shortened at ~ 10 Pa – a cross-over point at which G' and G'' had the same values (Fig 3e). As expected for a gel form, G' was greater than G'' at lower values, but lower than G'' in the stress region (>10 Pa) where transition from the gel into a liquid form occurred. Consistent with the classical shear-thinning behaviour found for the gels (Supplementary Fig 9b), which is also observed upon increasing shear in peptide, polysaccharide, and polymer hydrogels exhibiting dynamic fibrous, cross-linked, or reversible covalent networks,⁵⁰⁻⁵³ this finding indicates that the dense fibrous microstructure of SaBeH effectively supports the flow of SaBeH gels when stress is applied.

Point 20 – “MTT assay at different concentrations spanning log scale is needed.”

Response – presumably the point relates to the need for complementary cell viability assays. Therefore, additional tests and assays were performed including metabolic (MTT with a log of MTT concentrations), cytotoxicity (LDH release), proliferation by cell division (EdU,) metabolic long-term dye retention (CellTrace Violet) and distance tracking imaging analysis, including measurements done by flow cytometry. Please refer to the new Supplementary Figures 10 and 11e,f, and Movies 1 and 2. The corresponding text was modified as per our response to point 19 by the Reviewer.

Point 21 – “The space between unit and number is needed throughout the manuscripts (e.g., 25 °C).”

Response – as advised, the space between unit and number was checked throughout the manuscript and corrected where appropriate.

Reviewer 5:

This Reviewer has co-reviewed with one of the Reviewers above. No response is required.

Point-by-point responses to the Editorial and Reviewers' comments

Editorial

The manuscript has been revised as per the Editorial and Reviewers' comments. Specific changes in the main body of the manuscript are highlighted by a yellow background in an annotated pdf file "Related manuscript file".

Reviewer 2:

This Reviewer has raised additional points requiring clarification for some aspects of the study, which we have provided as follows:

Point 1 – *"In the respond the authors indicated that they performed three 100 ns replicas, and extended one of the simulations to 200 ns. However, this is not comment on the main manuscript, and can led to misunderstanding, as, for example, Fig. 1 represented a 100 ns MD frame, whereas page 6, line 128, references the 200 ns. It would be more suitable to include the 200 ns frame in Fig 1 or provide a justification for using the 100 ns frame instead. Additionally, in the RMSD plot provided in the supporting information (Fig. 5), a slight increase in RMSD is observed after 125 ns. It would be helpful to address whether this increase is a global trend, or if specific parts of the system are contributing more significantly to it."*

Response – to avoid confusion, in the revision Supplementary Figures 2-4 were modified to include distance maps and snapshots for both 100-ns and 200-ns simulations. The results for 100-ns and 200-ns simulations were practically identical. Therefore, the original 100 ns snapshots were kept in Fig 1 and references to Supplementary Figures 2-4 for 200-ns simulations were made in the corresponding text.

Also, as advised, the text was revised to addresses the peaking noted in the RMSD plots as follows:

Some peaking within 0.5-0.75 Å was observed in the RMSD plots at ~27 ns and 115 ns for 100-ns and 200-ns simulations, respectively. The peaking was consistent with the vacillation of the terminal prolines resulting in the varied spacings in the distance maps and was deemed marginal to reflect a more global variation in the assembly (Supplementary Figs 3 & 6).

Point 2 – *"Regarding the yield of the peptides, while the yield prior to purification has been included, the purity of the peptides should also be reported, both before and after purification (the yield will also be of interest)."*

Response – as advised, the purity and yields before and after purification were added in the corresponding Methods section in the revision.

Point 3 – *"The clarification regarding the three replicas and their similar behavior is appreciated. However, it would be valuable to include a plot over time of key distances or other relevant parameters, to highlight the reproducibility of the simulations."*

Response – as advised, an additional plot was included in the revision (new Supplementary Figure 5) showing distances between C α atoms of representative amino-acid pairs for 100-ns and 200-ns simulations. Please note that, as in Supplementary Figures 2-4, every distance represents the mean with a standard deviation of three replicas as indicated in the Figure captions and the Methods section.

Reviewer 3:

This Reviewer has offered additional points to clarify certain aspects of the study, which we have addressed as follows:

Point 1 – *"Cell migration. The authors have demonstrated cell migration in Supplementary Movies 1 and 2; however, the quantification provided remains inadequate. The travel distance per cell as shown in Supplementary Figure 11e is insufficient for a thorough analysis. It is essential that the authors include additional metrics such as the average migration speed and mean squared displacement. The videos depict minimal cell movement, which raises questions about the reported average travel distance"*

of approximately 30 μm per cell. This reviewer is unclear how such a travel distance corresponds to the minimal movement observed in the videos. Additionally, although the authors claim migration through lamellipodia or filopodia, the current videos and images do not robustly support this. Figure 4g, which displays filopodia morphology, is only a still image and does not convincingly demonstrate active migration. Cells displaying similar morphologies could be stationary and might employ different modes of migration. Therefore, the authors should provide close-up imaging of actively migrating cells to effectively illustrate this behavior.”

Response – as advised, additional metrics were provided to support the migration dynamics of the cells in the gels, including total displacement per cells, cell trajectory maps, MSD profiles, persistence and velocity – the new Supplementary Figure 13. Also, as requested, additional close-up images of individual cells were included in the revision – the new Figure 4i and Supplementary Figure 14.

Please also note that filopodia and lamellipodia are discussed in the context of cell responses to the SaBeH gels, manifesting in multipolar cell morphologies, which were in contrast to more polarised cell morphologies observed in native collagen matrices, and that cell migration is not exclusively defined by a unidirectional movement of a cell but is a complex process incorporating several events.

Multiple filopodia and lamellipodia support cell attachment, cell-cell interactions leading to spheroid formation, and cell migration. The cells with such morphologies can indeed be stationary and may employ different or mixed modes of migration, e.g., through blebbing. Furthermore, multiple protrusions help cells lower their membrane tension, reducing cell polarity and directionality of movement. When the membrane tension increases, multiple protrusions are suppressed in favour of a main polarised protrusion which sets up a streamlined movement. This scenario was observed in individual cells grown in SaBeH gels: multiple filopodia form and continuously re-arrange until a dominating protrusion emerges (new Figure 4i and Supplementary Figure 14). Profound membrane blebbing can then ensue supporting an amoeba-like behaviour. Thus, the results suggest an interplay between the stellate morphology of the cells, which maintain multiple protrusions in SaBeH, and the evolution of the protrusions into blebbing, which determines the anomalous diffusion of the cells in SaBeH gels via low-adhesion migration modes.

To clarify these points the corresponding text, with new supporting Refs: 60-67, was modified to read:

Cell migration is often heterogeneous and exhibits diffusion patterns that are anomalous to the Brownian motion (Supplementary Fig 13c).^{60, 61} Such patterns contain both directionality, which is expressed in linear migration trajectories, and tumbling, which is expressed in random trajectories.⁶²⁻⁶⁵ Cell tracks in SaBeH gels measured individually gave broad distributions of displacements per cell (Supplementary Fig 13b), but together coalesced into anisotropic patterns, which were more apparent for longer timescales (Supplementary Fig 13d). The ensemble averaged analysis of these trajectories, such as the mean-squared displacement (MSD), which averages cell positions and the ensemble of their migration trajectories, confirmed an anomalous cell diffusion in the SaBeH gels (Supplementary Fig 13e).⁶⁰⁻⁶⁵ The MSD curves appeared to be non-linear and consistent with superdiffusivity patterns suggesting a degree of persistence in migration trajectories.⁶²⁻⁶⁵ Indeed, this was observed. Persistence, expressed as a ratio of a net displacement versus a total distance travelled by a cell,⁶⁵ was characteristic of a directional migration, which appeared to increase over time (Supplementary Fig 13f). This tendency was reversed to that of velocity, which decreased over time (Supplementary Fig 13g), indicating that cell migration became more directional, but slower.

The observed migration dynamics were comparable to those described by others in 2D and 3D for different cell types including actively migrating cells such as T-cells, astrocytes, leukocytes and dendritic cells.⁶⁰⁻⁶⁵ Individually net distances travelled by cells in the SaBeH gels in the first hours were shorter suggesting some preference for comparatively more confined movements. Since persistence is determined by the ability of a cell to build a network of focal adhesions in the extracellular matrix, such a behaviour is reflective of the stellate polarity of cells in SaBeH gels. Multiple filopodia and lamellipodia lower the membrane tension of a cell, reducing its polarity and the directionality of movement.⁶⁶ When the membrane tension increases, multiple protrusions are suppressed in favour of a main polarised protrusion which sets up a streamlined movement.⁶⁷ This scenario can be observed in

individual cells grown in SaBeH gels: multiple filopodia form and continuously re-arrange until a dominating protrusion emerges (Fig 4i and Supplementary Fig 14a). Profound membrane blebbing can then ensue supporting an amoeba-like behaviour that is typical of cancer cells (Supplementary Fig 14b),⁶⁸ which employ blebbing for locomotion within their extracellular environment and extrusion from epithelia prior to metastasis.⁶⁹ Blebbing allows cells to release intracellular pressure by forming polarised protrusions – lobopodia at a leading end of the cell and uropodia at the rear end.⁶⁹ The cell mass moves towards lobopodia, whilst adhesive bonds between uropodia and the ECM are broken thereby promoting de-adhesion. Because lobopodia protrude faster than they can adhere to the ECM, the cell can move forward. Primary cells, such as HDFs, use similar mechanisms for migration by switching between polarised protrusions of low and high pressure which leads to their bi- or multi-polar shapes, as was observed in SaBeH gells.⁷⁰ Taken together the results suggest an interplay between the stellate morphology of the cells, which maintain multiple protrusions in SaBeH, and the evolution of the protrusions into blebbing, which determines the anomalous diffusion of the cells in SaBeH gels via low-adhesion migration modes. Thus, the results support the view of the current models that the dynamics of cellular migration is executed at complexity levels that is well beyond the individual components of cell migration.⁶⁰⁻⁶⁷

Point 2 – “Cell-matrix sensing. Figure 4a demonstrates that cells can indeed interact with SaBeH. However, the mechanism or binding site through which cells adhere to this scaffold remains unclear. The authors are encouraged to discuss how cells directly bind to the scaffold. Additionally, the statement, “Since SaBeH shares physical properties with other viscoelastic cell-supporting hydrogels based on native ECM proteins such as NCTI, synthetic peptides, polysaccharides, and polymers,⁴⁶⁻⁵³ the observed cell responses cannot be merely due to the bulk properties of the scaffold such as stiffness or viscoelasticity,” is overly simplistic. It is important to acknowledge that, beyond physical properties, the biochemical properties of SaBeH can significantly influence cellular responses. Therefore, it is unclear whether the observed responses are driven by the physical/morphological properties (e.g., the carpet-like micro-/nano-structure of SaBeH) or its biochemical characteristics. A discussion on this distinction would be valuable.”

Response – as encouraged, we expanded the discussion about the nature of cell-scaffold interactions in the revision, acknowledging also the complexity of properties that may influence cell responses. Similar to cell migration, the cell-scaffold interactions are indeed complex phenomena. It should also be re-emphasised that since SaBeH does not contain known biological cues, binding sites, adhesion motifs or recognition patterns, which would define biochemical characteristics, the role of such characteristics cannot be discussed unambiguously. Encouragingly, however, the lack of biology in SaBeH creates an opportunity for its tuneability.

To this effect the corresponding text, with new supporting Refs: 60, 65, 67, was modified to read:

Unlike NCTI, SaBeH does not contain known cell adhesion motifs, binding sites or receptor ligands. Therefore, in the absence of established biology, the basis of cell responses to the SaBeH scaffold must be due to its physical or morphological properties. Since SaBeH shares physical properties with other viscoelastic cell-supporting hydrogels based on native ECM proteins such as NCTI, synthetic peptides, polysaccharides, and polymers,⁴⁶⁻⁵³ the observed cell responses cannot be merely due to the bulk properties of the scaffold such as stiffness or viscoelasticity. Differences in cell behaviour are manifested in clear changes in the morphologies of the same cells seeded in NCTI and SaBeH and hence derive from cell interactions with the microstructure of the scaffold, i.e., the dense, carpet-like matrix of SaBeH versus long-range networks of NCTI (Fig 4e, f).

and

In this vein, SaBeH reduces the membrane tension of adhering cells while promoting their growth but lacks specific biological cues that could streamline the trajectories of cell migration.^{60, 65} Because SaBeH forms a dense, carpet-like matrix of chemically identical nanoscale cylinders, which cells profusely attach to, any cues that can straighten the directionality of travel should be in low abundance. This creates the potential to tune the design. By incorporating site-specific biological signals or binding

sites in the SaBeH matrix as a whole it should be possible to tailor the cell migration dynamics in favour of preferentially unidirectional or confined modes. With no known and pre-defined biology in the reported SaBeH design, the observed cell adhesion and migration may not be specified by biological cues. Unless SaBeH incorporates effective, yet unknown signals, cell migration in SaBeH gels is more likely to derive from the physical properties of the scaffold whose carpet-like microstructure provides virtually an unlimited space for focal adhesions (Fig 4e,f). Given that the scaffold has no persistent networks, the focal adhesions formed are short-lived but readily re-arrange, which necessitates their abundant formation manifesting in multipolar cell morphologies, promoting amoeboid modes of cell migration.⁶⁷⁻⁷⁰

Point 3 – “Scaffold morphology. The authors claim that the structural properties of the scaffold influence cell behavior. If this is the case, how does the scaffold structure change with gel concentration? In Supplementary Figure 9, stiffness increases nearly threefold when the concentration increases from 2% to 5%. Does this change result in a denser packing of the scaffold, or does it alter the structure in other ways? Using SEM imaging to examine how scaffold morphology changes with concentration would help clarify whether and how structural and mechanical properties are interrelated.”

Response – in the absence of known biology, the structural and morphological properties of the scaffold are bound to define the cell behaviour in the scaffold, incl. cell morphology and migration dynamics. The scaffold exhibits a carpet-like density at 1% (w/v), which remained largely comparable at 2% and 5% (w/v) concentrations, as gauged by electron microscopy – new Supplementary Figure 10a shows the gels at 2% and 5%, imaged as is. When imaged as microtomed (Supplementary Figure 10b), 5% gels appeared to have higher fibrous densities, which is indeed consistent with the marginally increased G' and $\tan \delta$ loss decreasing at higher frequencies for these gels.

To reflect on these points, the corresponding text was modified to read:

Since microtomes are 2D slices of 3D materials embedded in resin, the assemblies are expected to orient at varied angles with respect to the plane of the image, as observed (Fig 3c). As the density of the nanostructures increases, the impact of their relative orientation in the microtome images becomes less apparent. When imaged as is, the material exhibited comparable density at all gelation concentrations (Supplementary Fig 10a), while microtomed preparations of the material revealed relatively higher densities at higher gelation concentrations (2% and 5%, w/v) (Supplementary Fig 10b).

and

The same behaviour held true for 5 % gels, for which marginally increased G' (≥ 1 kPa) and $\tan \delta$ loss decreasing at higher frequencies (>60 rad/s) suggest increased fibrous density, which was confirmed in the microtomed preparations (Supplementary Fig 10b, c).

Reviewer 4:

This Reviewer has supported the manuscript as suitable for publication, and suggested one minor point, which we have addressed as follows:

Point 1 – “I would like to suggest that the authors add the units in the Abs spectrum of Figure 2c Please note that in the peptide purification protocol the wording buffer should be replaced by the word solution, because they are not buffers.”

Response – as advised, the corrections were made in the revision.

Reviewer 5:

This Reviewer has co-reviewed with one of the Reviewers above. No response is required.